# Effect direction meta-analysis of GWAS identifies extreme, prevalent and shared pleiotropy in a large mammal

Ruidong Xiang[1,2 ✉], Irene van den Berg[1,2], Iona M. MacLeod[2], Hans D. Daetwyler[2,3] & Michael E. Goddard[1,2]

In genome-wide association studies (GWAS), variants showing consistent effect directions across populations are considered as true discoveries. We model this information in an Effect Direction MEta-analysis (EDME) to quantify pleiotropy using GWAS of 34 Cholesky-decorrelated traits in 44,000+ cattle with sequence variants. The effect-direction agreement between independent bull and cow datasets was used to quantify the false discovery rate by effect direction (FDRed) and the number of affected traits for prioritised variants. Variants with multi-trait $p < 1e-6$ affected 1~22 traits with an average of 10 traits. EDME assigns pleiotropic variants to each trait which informs the biology behind complex traits. New pleiotropic loci are identified, including signals from the cattle *FTO* locus mirroring its bystander effects on human obesity. When validated in the 1000-Bull Genome database, the prioritized pleiotropic variants consistently predicted expected phenotypic differences between dairy and beef cattle. EDME provides robust approaches to control GWAS FDR and quantify pleiotropy.

[1] Faculty of Veterinary & Agricultural Science, The University of Melbourne, Parkville 3052 Victoria, Australia. [2] Agriculture Victoria, AgriBio, Centre for AgriBiosciences, Bundoora, Victoria 3083, Australia. [3] School of Applied Systems Biology, La Trobe University, Bundoora, Victoria 3083, Australia. ✉email: ruidong.xiang@unimelb.edu.au

Genome-wide association studies (GWAS) find mutations associated with complex traits. A GWAS produces estimates of the effect of each variant on a trait measured by a regression coefficient $b$ and its standard error (se). From this, a $p$ value can be calculated for the null hypothesis that the variant has no association with the trait. Using the $p$ values and the number of significant variants, it is common to calculate a false-discovery rate (FDR)[1]. However, this conventional FDR depends on the calibration of the $p$ value which in turn depends on the data conforming to the statistical model assumed in calculating the $p$ value.

The important implication of FDR is that true discoveries would be confirmed in a powerful, independent experiment. In this paper we directly use the ability to confirm a finding in an independent study to estimate the FDR by using the proportion of sequence variants which have an effect in the same direction in two independent datasets. We call this FDRed for 'effect direction' based on a statistical model we call Effect Direction MEta-analysis (EDME) for effect direction meta-analysis. EDME is easily extended to calculate the FDR by effect directions (FDRed) for a multi-trait GWAS. This will allow us to know the number of traits that a significant variant is associated with and precisely which traits these are.

Due to the recent availability of many GWAS results with large sample sizes in humans[2,3], meta-analysis of public GWAS summary statistics significantly improved our understanding of pleiotropy[4–8]. Naturally, traits that are genetically correlated must share part of their causal variants, however, if traits are uncorrelated, can we still detect widespread pleiotropy?

Our study starts with multi-trait GWAS of individual data where the traits are decorrelated by a Cholesky transformation[9]. After GWAS, we focus on using the information of the consistency of variant effect directions to identify pleiotropic variants associated with uncorrelated traits, instead of distinguishing different types of pleiotropy[10]. We aim to use: (1) the consistency of variant effects in different populations to evaluate the FDRed independent of their $p$ values; (2) the information on the consistency of variant effects for multiple traits across different populations to quantify pleiotropy at both the variant and trait levels. We propose an EDME of GWAS of 34 complex traits of over 44,000 dairy bulls and cows with over 17.6 million sequence variants. A validation analysis with the 1000-bull genome database confirms the informativeness of pleiotropic variants prioritised by the EDME model.

## Results

### Single-trait GWAS and conventional multi-trait meta-analysis in bull and cow populations.
The phenotype of the cows was based on their own record for each trait corrected for fixed effects, while that of the bulls was based on their daughter's performance. The phenotype correlation matrix among the 34 traits was calculated separately in the bulls and cows and an average correlated matrix was derived. This average correlation matrix was used to calculate a Cholesky factor (L matrix)[9] to be applied to the 34 traits. The Cholesky transformation decorrelates the 34 traits by ordering the traits from 1 to 34 and correcting each trait for all the traits before it in the list. We use this method because the traits vary in their completeness, i.e., some traits have more records than others. By ordering the traits from the trait with most complete data to the trait with least complete data, we can make use of all the data because the Cholesky transformation of the $K$th trait in the list only needs data on the $K-1$ traits and individuals with trait $K$ to have data on all $K-1$ traits. Thus the Cholesky transformation produces uncorrelated traits and makes maximal use of incomplete data (Supplementary Data 1,

Supplementary Figs. 1 and 2). The average relatedness between the bull and cow populations was around 0 (Supplementary Fig. 3). The transformation produced traits which were almost uncorrelated, but the correlations were not exactly zero in either the cow or bull populations, because the correlation matrices differed slightly between the sexes (Supplementary Fig. 2). The variance of each trait was close to 1 (Supplementary Data 1). For clarity in what follows, when describing results specifically related to Cholesky-transformed traits, a label 'CT' was used. For example, a trait called CT temperament was interpreted as the eighth trait in the list corrected for its preceding seven traits. When describing results related to original traits that were not Cholesky-transformed, a label 'raw' was used, e.g., raw temperament. The raw protein yield (prot, the first trait) was identical to the CT protein yield.

On average across 34 CT traits, the heritability was 0.42 (±0.035, standard deviation) in bulls and 0.164 (±0.029) in cows (Supplementary Data 1), which was expected as bull phenotypes were more accurate[11]. On average at $p < 1e−6$, 1900 (±813) variants were significant per trait in bulls and 2018 (±795) in cows. The FDR calculated by conventional methods (Bolormaa et al.[1]) at $p = 1e−6$ was 0.22 (±0.045) for bulls and 0.1 (±0.021) for cows across 34 CT traits.

Using the established meta-analysis method of ref. [1], a multi-trait GWAS was calculated within the bulls and cows separately (Supplementary Fig. 4a, b). Overall, many significant variants (associated with at least one CT trait) detected in one sex were significant in the other sex (Supplementary Fig. 4c and Supplementary Data 2). At the $p < 1e−6$ level, 25,454 variants were significant in bulls, 31,076 in cows and 14,587 in both bulls and cows. A weighted meta-analysis combining variant $t$ value from bulls and cows further increased the number of significant variants to 93,513 (Fig. 1 and Supplementary Data 2). The conventional FDR for all types of meta-analysis was smaller than 0.1% (Supplementary Data 3). A variant clustering analysis supports the existence of pleiotropic QTL distinguished from linkage disequilibrium (LD) between sequence variants (see Methods and Supplementary Fig. 5).

**The EDME model.** We introduce the EDME model to analyse GWAS summary statistics for 34 CT traits estimated from the bull and cow populations separately. In the EDME model, the trait-associated variants were categorised as: 'true negative' ($T_−$) variants that truly decrease the trait value, the 'true positive' ($T_+$) variants that truly increase the trait value and the 'true zero' ($T_0$) variants with no reliable effects on the trait (the left panel of Fig. 2a). However, the observed effect of each variant on a trait was classified as positive or negative so that when the effect of a variant in both bulls and cows was described it fell into one of the classes '++', '−−', '+−' and '−+' (Fig. 2a and Methods).

We assume that true positives were always observed in the ++ class and true negatives in the −− class and that true zero effect variants have one-fourth probability of being in any of the four classes. This then allowed the calculations

$$T_0 = 2 \times n(A), \tag{1}$$

$$T_− + T_+ = T_r = n(B) - \frac{T_0}{2}, \tag{2}$$

$$\text{Single-trait FDRed} = \frac{T_0}{n(A) + n(B)}, \tag{3}$$

$$\text{Single-trait TDRed} = \frac{T_r}{n(A) + n(B)}. \tag{4}$$

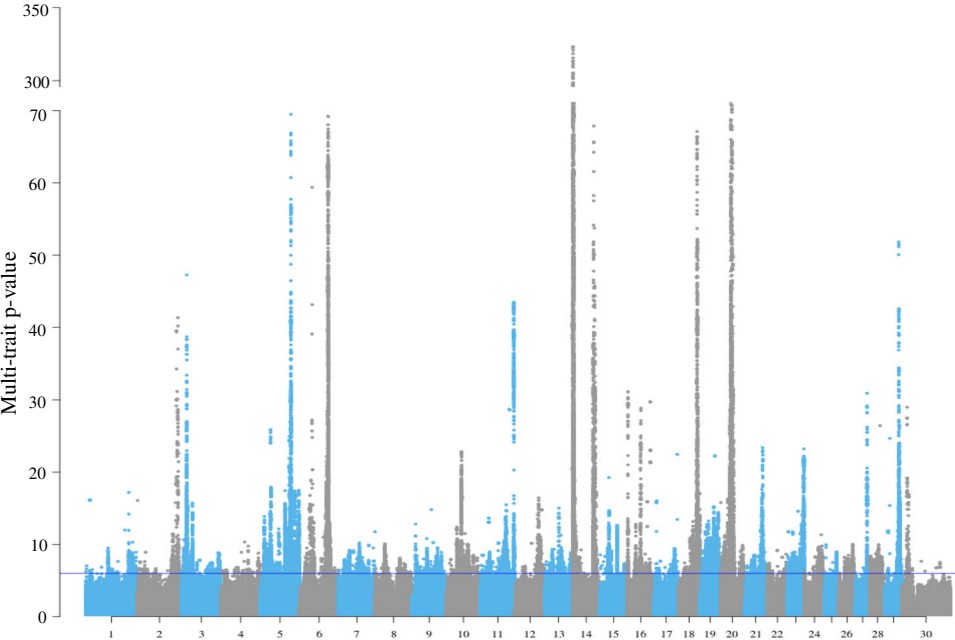

**Fig. 1 The Manhattan plot of *p* values of weighted meta-analysis combining GWAS results from bulls and cows for each CT trait.** The blue line indicates the weighted multi-trait meta-analysis *p* value = 1e−6.

Where $n(A)$ was the observed number of variants with inconsistent effect directions ('+−' and '−+') and $n(B)$ was the observed number of variants with consistent effect directions ('++' and '−−') in GWAS of two different populations. $T_r$ denoted the combined number of $T_-$ and $T_+$. Then, the FDR and TDR by effect directions (single-trait FDRed and TDRed, respectively) were calculated as Eqs. 3 and 4.

**The single-trait EDME and FDRed**. Based on the number of variants with inconsistent effect directions between sexes ($n(A)$), the FDRed and conventional FDR averaged across 34 CT traits were compared within a range of $p$ value thresholds imposed in both sexes (Fig. 2b). Across 34 CT traits, the number of variants significant in both sexes ranged from 241,657(±6071) at $p = 0.1$, 798(±424) at $p = 1e−6$, to 14(±12) at $p = 1e−100$ (Fig. 2b). Both types of FDR decreased as the $p$ value threshold was made more stringent, with the FDRed being smaller than the conventional FDR. At the $p = 1e−3$ level, amongst 2850(±1218) significant variants for both sexes, the single-trait FDRed was 0.032(±0.015) while the conventional FDR was 0.98(±0.015). At the $p = 1e−9$ level, amongst 408(±213) significant variants for both sexes, the single-trait FDRed dropped to 0 and the conventional FDR was 1.6e−05(±3.4e−06) (Fig. 2b). When imposing the $p$ value threshold only in one sex, the FDRed was on average higher than the conventional FDR until a very small $p$ threshold was achieved (Supplementary Data 4). This may be due to differences in LD structure between cows and bulls and/or to the lack of power to detect real effects in both sexes. However, as shown above, if the consideration was restricted to sequence variants that were significant in both sexes, the agreement in the sign of the effect was high and hence the FDRed was low.

**The multi-trait EDME**. The single-trait EDME was extended to multiple traits (see Methods) to calculate FDRed and quantify pleiotropy. For each variant, the dot product of the vector of signed *t* values of both sexes ($\Pi$, Eq. 5, detailed in below) was used to quantify the overall agreement of the effect directions on multiple traits per variant. A positive $\Pi$ indicates effects in the

same direction between bulls and cows, whereas a negative $\Pi$ indicates the opposite direction of effect between bulls and cows.

$$\Pi_i = t_{i1} \cdot t_{i2}^T, \qquad (5)$$

$$\text{Multi-trait FDRed} = [n(\text{variant}|\Pi_i \leq 0) \times 2]/n(\text{total}), \qquad (6)$$

$$\text{Estimated number of TE} = 2 \times \sum_1^K n(B)_{\text{MTR}} - K, \qquad (7)$$

$$\text{Number of TC}_{Tr \times V} = n(\text{variant}) \times \overline{\text{TE}}. \qquad (8)$$

For Eq. 5, $t_{i1}$ was the *t* value vector for variant$_i$ of population 1 with $K$ elements and $t_{i2}$ was the *t* value vector for variant$_i$ of population 2 with $K$ elements; $K$ was the total number of traits analysed in each population ($K = 34$ in this study). For a group of variants, $\Pi$ values allowed the use of the same logic calculating the single-trait FDRed (Eqs. 1 and 3) to calculate a multi-trait FDRed (Eq. 6). For Eq. 6, $n$(total) was the total number of variants selected.

As well as this multi-trait estimate of FDR, it is also possible, for each sequence variant, to count the number of traits with the same direction of effect in population 1 (bulls) and population 2 (cows). Allowing for the fact that half the traits were expected to be in the same direction by chance, the EDME logic can estimate the number of traits showing a True Effect (TE) for each variant (Eq. 7). While this estimate was subject to large sampling error for any variant, its average across a large group of variants was consistent (as shown in the results described in the following). For Eq. 7, $\sum_1^K n(B)_{\text{MTR}}$ was a natural number ranging from 0 to $K$, where for each trait (out of $K$ traits), $n(B)_{\text{MTR}} = 1$ if the variant has the same effect direction and $n(B)_{\text{MTR}} = 0$ if the variant has different effect directions between the two populations.

The estimated number of TE averaged across a group of variants ($\overline{\text{TE}}$) × the number of variants in that group ($n$(variant), Eq. 8) estimated the number of true combinations (TC) of trait by variant effects, $\text{TC}_{Tr \times V}$. $\text{TC}_{Tr \times V}$ can be used as a cut-off to prioritise informative trait-variant combinations. Two additional pieces of information can be extracted from the prioritised trait-variant combinations: (i) the observed number of TE for each

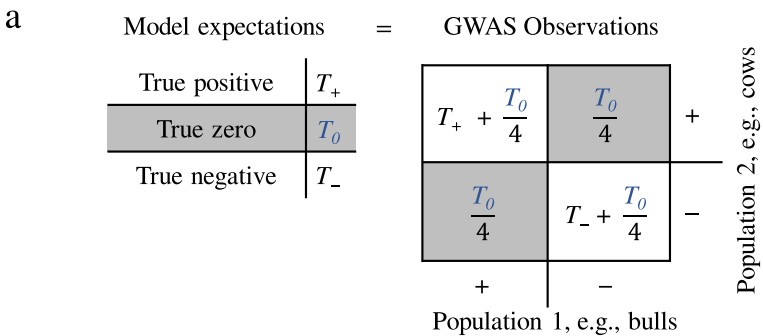

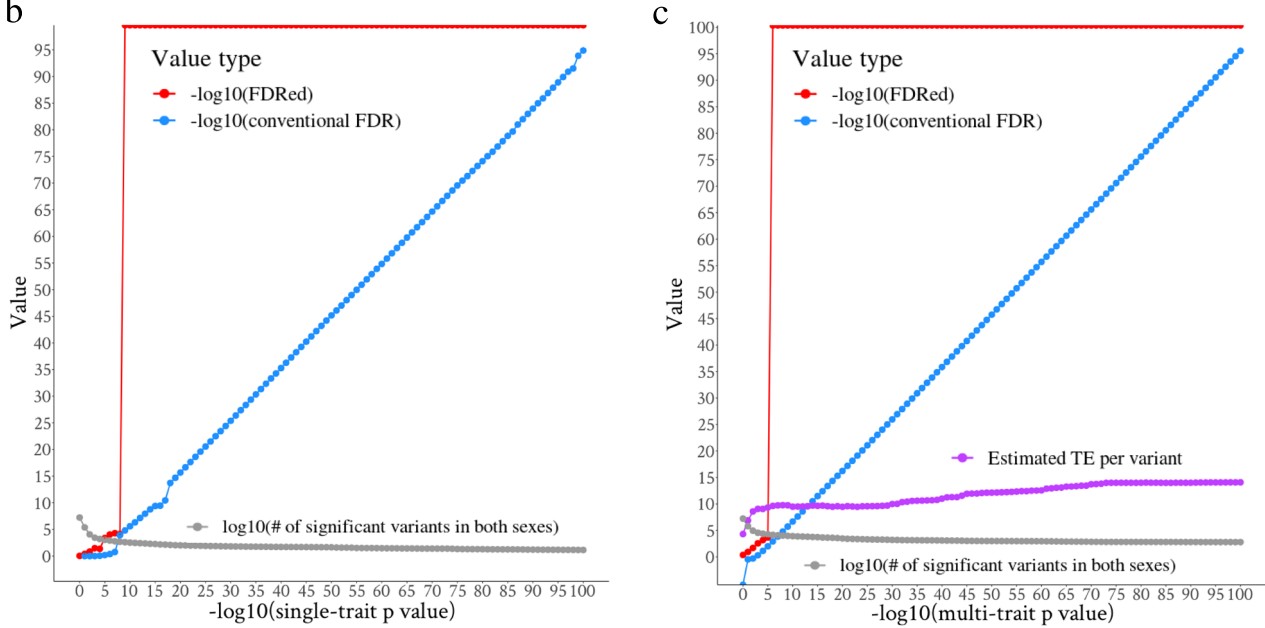

**Fig. 2 The graphical representation of the EDME model and false-discovery rate (FDR). a** The model expectations of true positive, zero and negatives in the left panel are equated with the observation of variant effect directions (the number of '++' in the first quadrant, '−+' in the second quadrant, '+−' in the third quadrant and '−−' in the fourth quadrant) of GWAS in two populations (e.g., bulls and cows). The grey boxes indicate the first solvable equation where the number of ('−+' and '+−') = $\frac{T_0}{2}$. **b** Distribution of FDR effect direction (FDRed) and the conventional FDR for a range of single-trait $p$ value thresholds imposed for both sexes. At each $p$ value point, the mean of FDR and its standard error across 34 CT traits were given. The single-trait FDRed was based on the proportion of variants with inconsistent effect directions between sexes (Eq. 3). The grey line indicates the number of variants significant in both sexes at each $p$ value threshold. **c** Distribution of FDRed and conventional FDR for a range of multi-trait $p$ value thresholds imposed for both sexes (grey line). The multi-trait FDRed was based on the dot product (Π) of the variant effects (beta/s.e.) from two sexes (Eq. 5). Using the count of variants with consistent effect directions across multiple traits, EDME estimates the number of true effects (TE) averaged across a selection of variants (Eq. 7).

variant suggesting the extent of pleiotropy at the variant level; (ii) the pleiotropic variants related to each trait, variants$_{trait}$, suggesting pleiotropic variants that affect each trait (illustrated in the following results).

**The genome-wide characteristics of multi-trait FDR and pleiotropy.** Similar to the single-trait FDRed described above, the multi-trait FDRed remained smaller than the conventional FDR when imposing the multi-trait $p$ value threshold in both sexes (Fig. 2c). The number of variants significant in both sexes ranged from 641,399 at multi-trait $p = 0.1$, 14,587 at $p = 1e−6$, to 631 at $p = 1e−100$ (Fig. 2b). At the multi-trait $p = 1e−2$ level, amongst 89,223 significant variants for both sexes, the multi-trait FDRed was 0.02 while the conventional FDR = 1. At $p = 1e−3$, the multi-trait FDRed was 0.003 (conventional FDR = 0.49) and it dropped to 0 at the $p = 1e−6$ level where the estimated number of TE per variant was 9.63. That is, the average number of traits affected by a single-nucleotide polymorphisms (SNP) in this

group is 9.63. Additional analysis with functional data[12,13] indicated that both single-trait and multi-trait FDRed can help find informative variants at lenient $p$ value thresholds where conventional FDR was large (Supplementary Note 1). When imposing a significance threshold in only one sex, at the multi-trait $p$ value threshold of 1e−6, the multi-trait FDRed was 0.14 in bulls and 0.2 in cows (Supplementary Data 5).

To further dissect the pleiotropy with a group of informative variants, 93,513 variants with the weighted multi-trait $p$ value ($p_{wm}$) < 1e−6 from the weighted meta-analysis combining GWAS results of bulls and cows (Fig. 1), were selected. Applying Eq. 8 to this group of variants, the number of TC of trait by variant ($TC_{Tr×v}$) was estimated as 93,513 × 9.63 (estimated $\overline{TE}$) = 900,820 at this $p$ threshold. This indicated that 900,820 combinations of trait by variant can be interpreted as 'true'. To specify these combinations, we chose those ranked as top 900,820 by their absolute weighted $t$ values combing sexes across 34 CT traits. These combinations of trait by variant contained 92,537

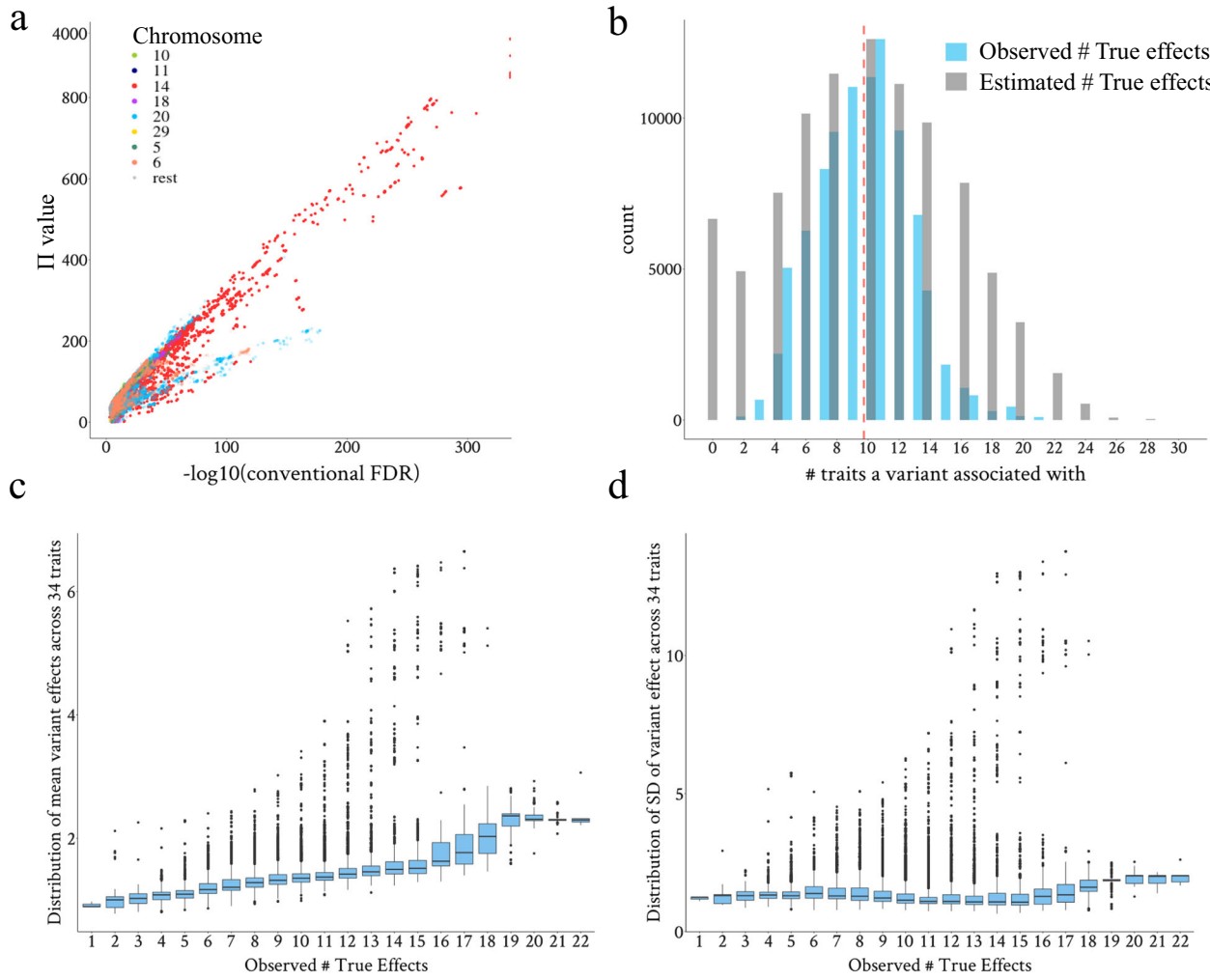

**Fig. 3 The distribution and characteristics of pleiotropic variants prioritised by the EDME model in combination with the weighted multi-trait _p_ value.**
**a** Scatter plot for the relationship between the variant Π value and the conventional FDR based on the weighted multi-trait meta-analysis _p_ value.
**b** Distribution of the estimated and observed number of true effects on traits for each variant. The estimated number (#) of true effects per variant is based on Eq. 5, and the observed number of true effects for each variant is based on the prioritised trait by variant combinations using Eq. 8. **c** Boxplots of averaged effect size (absolute weighted _t_ value of bulls and cows, _b_/se) across 34 CT traits for each one of the categories of variants with a varied number of traits truly affected. **d** Boxplots of standard deviation (SD) of effect size across 34 CT traits for each one of the categories of variants with a varied number of traits truly affected.

unique variants with Π > 0 (99% of 93,513 variants with $p_{wm} < 1e$ −6), and their Π values, which indicate the effect direction agreement for multi-trait analysis, were correlated with their conventional FDR based on $p_{wm}$ (Fig. 3a).

For each of the prioritised 92,537 variants, we counted the number of traits with which they were associated among the 'true' 900,820 combinations of trait by variant and this was defined as the observed number of TE. The observed number of TE for each variant ranged from 1 to 22 with a mean of 9.7 (Fig. 3b and Table 1). Also, these prioritised pleiotropic variants tagged up to 3550 (11.6% of 30,514) and 3958 (13% of 30,440) clusters representing different QTL in bulls and cows, respectively. Based on the prioritised pleiotropic variants that were also located within these clusters, each cluster had 9.79 and 9.71 (ranging from 1 to 22) observed $\overline{TE}$ in bulls and cows, respectively. These results were consistent with the average number of $\overline{TE}$ per variant (9.63) estimated previously (Eq. 7, Fig. 4b). The pleiotropic characteristics of these variants were detailed in Table 1. The $p_{wm}$ and conventional FDR decreased with the increase of the Π value which was indicative of the consistency of effect direction

between two populations. A list of the top 500 variants based on Π value within each category of pleiotropy was shown in Supplementary Data 6.

The most pleiotropic variants accounting for 0.00003% of the genome with observed TE on 22 CT traits were close to the _GC_ region (Chr6:88.8M) (Supplementary Data 6). Variants with TE on 22 CT traits also tagged the gene _ZNF613_ (Chr18:58.1M) which was close to the _CTU1_ region (Chr18:57.5M). Variants from the _CTU1_ region had observed TE on 20 CT traits. Both _GC_ and _CTU1_ loci were reported as lead dairy cattle pleiotropic loci in the 2017 pleiotropy study[9]. Other commonly known major loci for dairy cattle (see refs. [9,11,14–16]) were also captured, including _DGAT1_ (Chr14:1.8M) that had TE on up to 17 CT traits, _MGST1_ (Chr5:93.9M) that had TE on ~13 CT traits, _CSN1S1_ (Chr6:87.1M) that had TE on ~12 CT traits, _GHR_ (Chr20:32M) that had TE on ~15 CT traits and _ENSBTAG00000048091_ (beta-lactoglobulin-2-like), a potentially duplicated gene of the adjacent _PAEP_ (beta-lactoglobulin, Chr11:103.2M) that had TE on ~11 CT traits.

Less known pleiotropic loci were also identified. A highly pleiotropic loci _PC_ (Chr29:45.6M), coding for the enzyme

**Table 1 Characteristics of selected informative pleiotropic variants.**

| Observed TE type | # Variants | Genome fraction (%) | $p_{wm}$ (se) | Conventional FDR (se) | $\Pi$ (se) |
|---|---|---|---|---|---|
| 1 | 3 | 0.00002 | 1.0E−09(9.9E−10) | 1.1E−07(1.0E−07) | 20.7(0.9) |
| 2 | 116 | 0.0007 | 1.7E−07(2.5E−08) | 3.8E−05(1.4E−05) | 24.8(1.1) |
| 3 | 666 | 0.0038 | 1.1E−07(8.3E−09) | 4.4E−05(4.5E−06) | 30.1(0.5) |
| 4 | 2195 | 0.012 | 9.0E−08(4.0E−09) | 2.1E−05(1.8E−06) | 34.7(0.5) |
| 5 | 5040 | 0.029 | 1.2E−07(3.2E−09) | 1.5E−05(9.7E−07) | 36.3(0.5) |
| 6 | 6264 | 0.036 | 8.5E−08(2.4E−09) | 1.8E−05(9.2E−07) | 45(0.4) |
| 7 | 8323 | 0.047 | 1.2E−07(2.3E−09) | 2.0E−05(8.8E−07) | 48(0.4) |
| 8 | 9540 | 0.054 | 1.3E−07(2.5E−09) | 3.4E−05(1.4E−06) | 52.5(0.4) |
| 9 | 11,035 | 0.063 | 1.6E−07(2.5E−09) | 5.4E−05(2.1E−06) | 52.1(0.4) |
| 10 | 11,354 | 0.064 | 2.0E−07(2.8E−09) | 5.1E−05(1.6E−06) | 54.3(0.5) |
| 11 | 12,622 | 0.071 | 1.8E−07(2.4E−09) | 5.6E−05(1.7E−06) | 49.8(0.5) |
| 12 | 9589 | 0.054 | 1.4E−07(2.4E−09) | 5.8E−05(2.0E−06) | 63.2(1.2) |
| 13 | 6791 | 0.038 | 1.2E−07(2.7E-09) | 5.4E−05(2.5E−06) | 67(1.8) |
| 14 | 4289 | 0.024 | 1.1E−07(3.4E−09) | 5.1E−05(3.4E−06) | 168.2(6.7) |
| 15 | 1834 | 0.010 | 8.9E−08(4.6E−09) | 3.7E−05(3.4E−06) | 129.4(9.9) |
| 16 | 1072 | 0.006 | 5.5E−08(5.1E−09) | 1.9E−05(1.9E−06) | 160.3(14.1) |
| 17 | 817 | 0.0046 | 1.4E−08(1.5E−09) | 9.4E−06(8.8E−07) | 176.8(17.4) |
| 18 | 304 | 0.0017 | 8.1E−09(4.2E−09) | 5.0E−06(1.9E−06) | 120.7(10.3) |
| 19 | 448 | 0.0025 | 1.4E−10(6.2E−11) | 5.3E−07(2.2E−07) | 140(1.2) |
| 20 | 130 | 0.0007 | 3.9E−20(3.9E−20) | 2.9E−17(2.9E−17) | 144.8(1.5) |
| 21 | 99 | 0.0006 | 6.0E−25(4.2E−25) | 1.7E−22(1.2E−22) | 143(1.2) |
| 22 | 6 | 0.00003 | 2.4E−31(1.9E−31) | 8.8E−29(7.0E−29) | 155.4(18.8) |

For each pleiotropy category, i.e., the observed number of true effects (TE) for each variant, the number (#) of variants for each type of TE, the number of variants relative (genome fraction) to the total number sequence of variants (~17M), averaged weighted multi-trait $p$ value ($p_{wm}$), conventional false-discovery rate (FDR) and the dot product of $t$ value between sexes ($\Pi$, Eq. 5) are shown with their standard error (se) in parenthesis.

pyruvate carboxylase which is needed for gluconeogenesis and lipogenesis, had TE on ~18 CT traits. A pleiotropic locus with TE on ~12 CT traits was *SLC22A6* (Chr29:41.9M) which had reported roles in regulating water metabolism in mammals[17]. In addition, variants from the X chromosome (chromosome 30) with pleiotropic effects were also identified. Tagged genes included *CD40LG* (Chr30:20.2M) and *ARHGEF6* (Chr30:20.3M), although these variants only affected 2–3 CT traits (Supplementary Data 6).

Further analysis of those informative pleiotropic variants showed that variants affecting a larger number of CT traits had overall larger effect sizes across 34 CT traits, compared to those variants that affected a smaller number of CT traits (Fig. 3c). Many variants that showed widespread pleiotropic effects, i.e., TE > 8, had large effect size variation across 34 CT traits (e.g., the outliers in boxplots of Fig. 3d). This meant that these pleiotropic variants could have large effects on some traits but small effects on other traits. Some noticeable variants/loci associated with multiple CT trait groups were detailed in Supplementary Note 2, Supplementary Figure 6, Supplementary Datas 7 and 8.

**Characteristics of trait-related variants.** As described above, the 900,820 combinations of trait by variant assigned pleiotropic variants to each CT trait (variants_trait). Each set of variants_trait was tested for whether the variants_trait had specific associations with the trait that they were assigned to. For each trait, the single-trait FDRed was calculated using the trait-related variants_trait. Such single-trait FDRed (average < 0.1) was much smaller than the single-trait FDRed calculated using those general pleiotropic variants associated with any one of the 34 traits (multi-trait $p_{wm}$ < 1e−6, average single-trait FDRed > 0.7) (Supplementary Fig. 7a). Also, previously identified variants regulating cattle stature[18] had the strongest enrichment (adjusted $p$ = 1.9e–08) in the variants_trait related to dairy cattle CT stature (trait 19) (Supplementary Fig. 7b). These results support that the variants_trait had specific associations with the traits they were related to. However, we note

that these variants were related to CT traits and the interpretation of them is different from the raw traits.

**The sharing of pleiotropic variants between traits.** Among the 900,820 combinations of trait by variant, we counted the number of variants that affected each pair of CT traits (Fig. 4a). Many pairs of CT traits have higher counts than expected by chance. While many pleiotropic variants_trait were detected for CT milk production traits (the first four traits) and they had significant sharing of pleiotropic variants_trait amongst themselves, CT milk production traits did not have significant sharing of pleiotropic variants_trait with every other CT trait. Although traits like CT body condition score (BCS) had the least amount of pleiotropic variants_trait, CT BCS had significant sharing of pleiotropic variants_trait with a wide range of other CT traits.

For each CT trait, the pleiotropic variants_trait sharing index, i.e., the number of pleiotropic variants_trait overlapping divided by the number in the union for each trait pair, averaged across the other 33 traits was calculated. As shown in Fig. 4b, the average pleiotropic variants_trait sharing index was used to rank CT traits from 'influential' (sharing pleiotropic variants_trait with many traits) to 'independent' (sharing pleiotropic variants_trait with few traits). The first trait protein yield was ranked as the most influential trait, followed by other CT milk production traits. The CT type trait Bone (trait 21), the original trait of which indicated the flatness of legbone of the cattle but is negatively correlated with fatness, ranked the fifth for its influence. The last CT trait BCS ranked relatively high for its influence, given its small number of pleiotropic variants_trait. The CT temperament (trait 8) and the overall type (OType, trait 25) were ranked as relatively independent amongst traits analysed (Fig. 4b).

**Raw trait relationships informed by Cholesky decorrelation.** The aforementioned results showed that the CT milk production traits had limited sharing of pleiotropic variants_trait with other CT traits (Fig. 4a). The Cholesky transformation corrected each

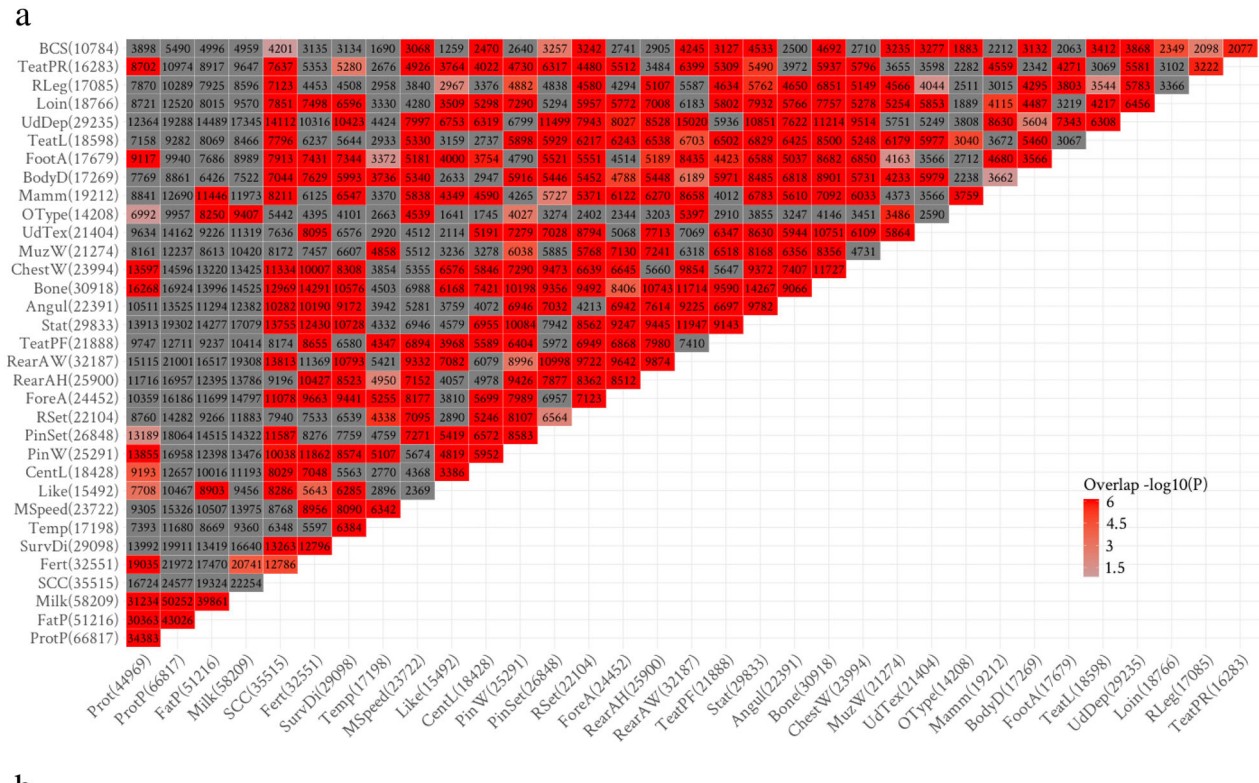

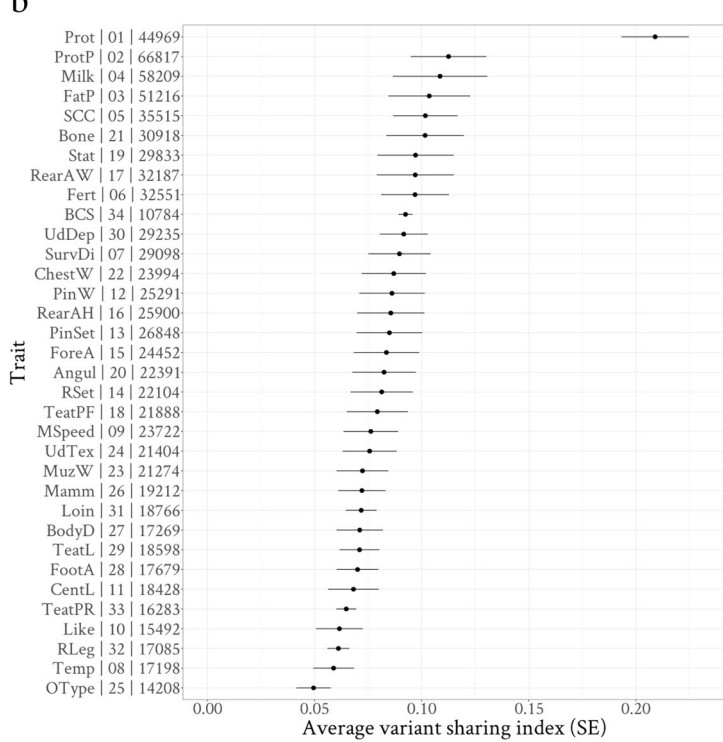

**Fig. 4 The sharing of pleiotropic variants between Cholesky-transformed (CT) traits. a** The sharing of pleiotropic variants related to each CT trait, variants$_{trait}$, between pairs of CT traits. The numbers in parentheses are the assigned pleiotropic variants$_{trait}$ for each CT trait based on the true combinations of trait by variant prioritised by EDME. The numbers in boxes are the number of variants in the overlap between pairs of CT traits. Red boxes indicate that the sharing of pleiotropic variants$_{trait}$ between CT traits is significantly different from random based on hypergeometric tests (accounting for the number of pleiotropic variants$_{trait}$ assigned for each trait and the total number of selected variants). **b** The ranking of CT traits using the sharing index of pleiotropic variants$_{trait}$, calculated as the number of overlap over the number of union for each CT trait pair, averaged across the other 33 traits. The y-axis has three columns separated by '|' that are defined from left to right as: the CT trait name, the order of the Cholesky transformation and the number of pleiotropic variants$_{trait}$ assigned to that CT trait. The x-axis indicates the average sharing index of pleiotropic variants$_{trait}$. Error bars indicates the standard error of the average of the sharing index of pleiotropic variants$_{trait}$ across 33 CT traits.

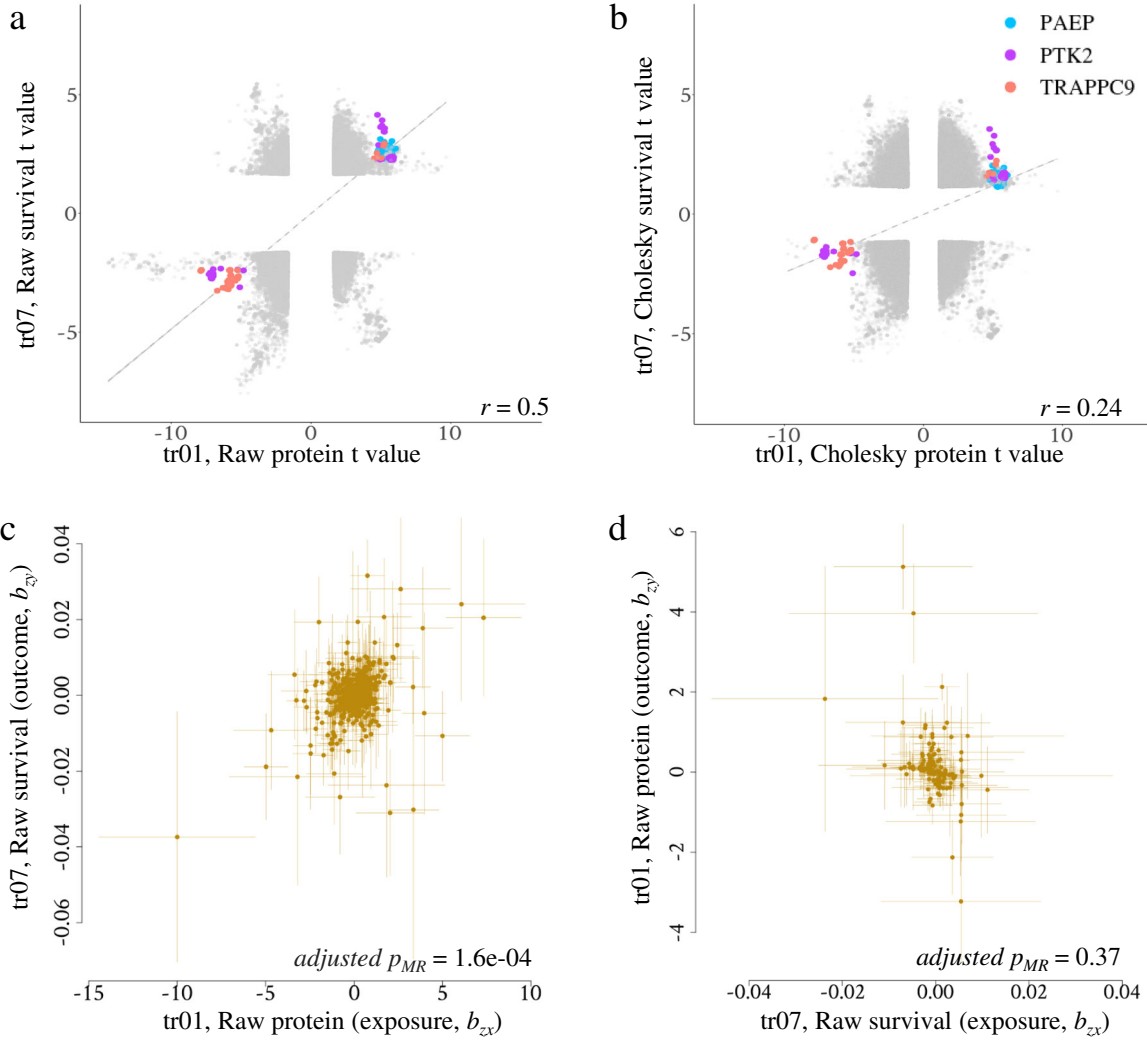

**Fig. 5 An example of the putative causal association between protein (trait 01) and raw survival (trait 07). a** scatter plot of GWAS $t$ ($b$/se) value of raw traits of protein and survival. Correlation ($r$) of effects between traits were indicated. **b** Scatter plot of GWAS $t$ ($b$/se) value of Cholesky traits of protein and survival. **c** Plot of effect sizes of forward Mendelian Randomisation using raw trait protein as exposure ($x$, causal variable), raw trait survival as outcome ($y$, effect variable) and non-pleiotropic variants as instrumental variables ($z$)[20]. Bonferroni multi-testing adjusted $p$ value of Mendelian Randomisation ($p_{MR}$) is indicated. **d** Plot of effect sizes of reverse Mendelian Randomisation using raw trait survival as the exposure and raw trait protein as the outcome.

target raw trait for all preceding raw traits. For instance, the CT survival (trait 7) was corrected for raw traits 1–6 which included the trait protein yield (trait 1). The correction used the overall regression coefficient. Therefore, a sequence variant that affected protein yield and raw survival by the amount predicted by the regression of raw survival on protein predicts, had no effect on the CT survival (trait 7). If all variants that affected raw protein yield had the predicted effects on raw survival, then none of them would have an effect on CT survival. This is indeed almost what we observe (Fig. 5a, b). The simplest interpretation of this pattern of effects is that these variants only affected raw survival because of their effects on protein yield. In other words, protein yield tends to cause raw survival, or farmers cull cows with low protein yield. Figure 5a, b highlights the effects of variants from key genes associated with the raw trait of protein yield and survival.

To verify this causal inference, bidirectional Mendelian Randomisations[19] (MR) using variants as instrumental variables implemented in GSMR[20] were conducted for 50 raw trait pairs that had the least amount of variant sharing after Cholesky transformation (Supplementary Data 9). Defined as the Bonferroni-correction

adjusted $p_{MR} < 0.05$ for forward MR and adjusted $p_{MR} \geq 0.05$ for reverse MR, 8 putative causal relationships were identified (Supplementary Data 9), including protein $\xrightarrow{\text{cause}}$ survival (Fig. 5c, d). Additional plots of the MR analysis indicated the existence of directional pleiotropy[20,21] in the MR of survival protein, but not in the MR of protein $\xrightarrow{\text{cause}}$ survival (Supplementary Fig. 8). Given that there were a total of 218 pairs with adjusted $p_{MR} < 0.05$ for forward MR and adjusted $p_{MR} \geq 0.05$ for reverse MR and that a total of 2664 MR tested, the enrichment of the 8 putative causal pairs out of 50 was small but different from chance ($p_{overlap} = 0.043$ based on hypergeometric test). Raw somatic cell count (SCC) also had some putative causal effects on raw mammary related traits (Supplementary Data 9).

**Pleiotropic variants**. To the best of our knowledge, we defined the novel pleiotropic variants as those which were outside the ±2 Mb region of the 1181 pleiotropic SNPs with $p_{wm} < 1e−6$ in the 2017 study[9] that used high-density SNP array genotypes to analyse 25 dairy cattle traits. These previously detected 1181 pleiotropic

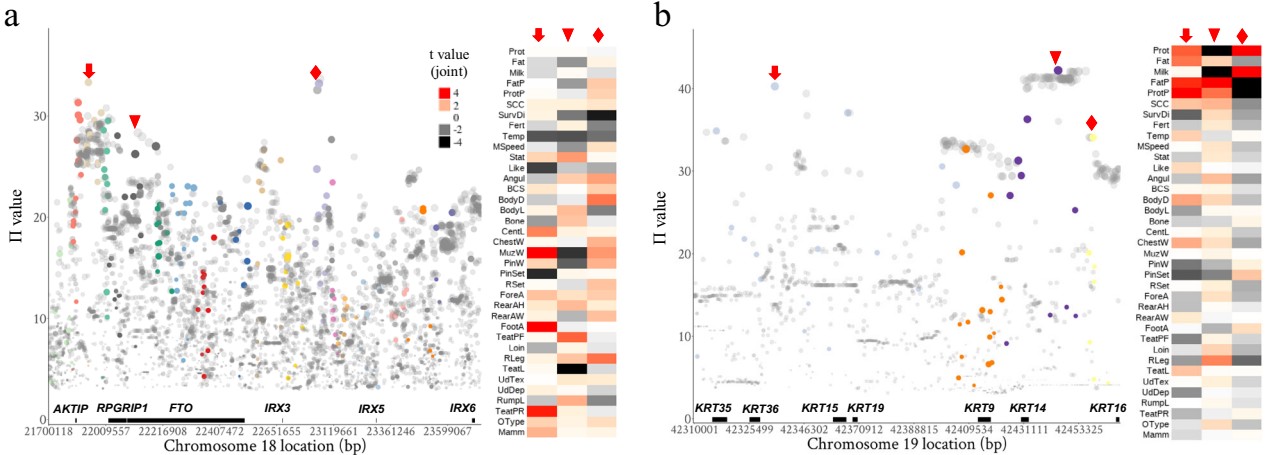

**Fig. 6 Examples of pleiotropic variants. a** The regional Manhattan plot for the pi value of the *AKTIP-FTO-IRX3-IRX5* locus together with the heatmap of effect sizes (*t* values, b/se) across 37 raw traits shown for top pleiotropic variants labelled by an arrow, triangle, and diamond. **b** The regional Manhattan plot for the pi value of the *KRT35-KRT14-KRT16* loci together with the heatmap of effect size across 37 raw traits shown for top pleiotropic variants labelled by an arrow, triangle, and diamond. In small heatmaps, the *t* values were calculated by fitting three targeted variants jointly for each raw trait (GCTA-COJO[37]). Larger dots had smaller *p* values of the weighted multi-trait meta-analysis. Dot colour represented the pruned variants from different clusters of variant pairs (see Methods).

SNPs tag 65,420 sequence variants out of the 92,537 informative pleiotropic variants identified in the current study. According to the annotation of Ensembl VEP[22], the remaining 27,117 pleiotropic variants were related to 1347 Ensemble genes and around 10% of these variants were not intergenic or intronic variants. An LD pruned set (LD $r^2 < 0.1$ in 2 Mb sliding windows) of these variants were detailed in Supplementary Data 10. These pleiotropic QTL included variants close (<0.5 Mb) to *IGFBP7* (Chr6:74.5 M)[23], *DNMT3B* (Chr13:62.6M)[24], *SLC24A4* (Chr21:57.6M)[25], *SLC37A1*[26] and *FTO* (Chr18:22.1M)[27] which were reported as single-trait or production trait QTL previously. However, there were also many loci not previously reported in cattle, such as intergenic variants tagging *MCF2* (ChrX:23M), *CTNNA2* (Chr11:54.9M), *NSF* (Chr19:46M) and *KRT14* (Chr19: 42.4M).

Two examples of the less reported pleiotropic loci in cattle are given in Fig. 6a, b and the variant effects on 37 original traits were detailed. On cattle chromosome 18 (22.5M), the top variant within the intronic region of *FTO* gene had pleiotropic effects on raw fat percentage, milk yield and many raw linear assessment traits, including likeability (Like), rear teat placement (TeatPR) and teat length (TeatL) (Fig. 6a). Variants with stronger pleiotropic effects than the *FTO* variants were found from the intergenic region between the adjacent *IRX3* and *IRX5* and between *AKTIP* and *RPGRIP1*. Variant clustering (see Methods) already indicated the existence of multiple QTL in this region (Fig. 6a). To further verify if these three variants mark more than one causal variant, we fitted all three simultaneously. Although close to *FTO*, all three variants were significant when fitted together and all three had different patterns of effects across raw traits (Fig. 6a). This observation appeared to mirror the bystander gene case of the human *FTO* where the regulatory effects of *FTO* on human body index and obesity risks are dependent on its neighbour genes[28–30]. All three loci had effects on raw temperature (Temp).

On chromosome 19, many pleiotropic variants were identified from the 42.2 to 42.4M region (Fig. 6b), tagging a cluster of keratin genes, usually making up the cornified surface of rumen and skin[31], including *KRT9*, *KRT14*, *KRT15*, *KRT16*, *KRT19*, *KRT35*, and *KRT36*. Again, cluster analysis indicated the existence of several QTL in this region. The joint analysis showed that selected top variants had a correlated pattern of effects on

milk production traits. These same variants had a less correlated pattern of effects on raw traits of SCC, temperament and many raw linear type traits of the mammary gland (Fig. 6b), although these effects were relatively small.

**Validating the pleiotropic variants prioritised by EDME and *p* value.** Three sets of pleiotropic variants were selected for validation analysis with 37 raw traits (see Methods and Supplementary Data 11): (1) the 92,537 informative pleiotropic variants prioritised by the EDME model after $p_{wm} < 1e-6$ selection, (2) 93,513 variants prioritised by $p_{wm} < 1e-6$ alone and (3) a random set of 93,513 variants. Overall, both the EDME with $p_{wm}$ and $p_{wm}$ alone prioritised pleiotropic variants predicted that dairy cattle were genetically better than the beef cattle for dairy traits and the difference in predictions between beef and dairy was generally lowest in the Random set (Fig. 7). Note that for raw SCC, fertility (Fert), temperament (Temp), milking speed (MSpeed) and likeability (Like), larger gEBV indicates poorer dairy trait performance. Based on our knowledge, larger gEBV in dairy than beef cattle is expected for 13 raw traits (Supplementary Data 12). For these 13 traits, variants prioritised by EDME combined with *p* value showed the most frequent (9/13 in bulls and 7/13 times in cows) prediction of the largest advantage in gEBV of dairy over beef cattle (Fig. 7) compared to other sets.

## Discussion

In this paper, we focus on finding pleiotropic variants and identifying the traits that each variant affects using EDME. Our approach differing from other published studies in three respects. Firstly, we use individual animal data rather than GWAS summaries. This is partly out of necessity because few summary data are available for cattle and most non-human species, but also, because some information is lost in summaries, it is better to start with individual data when this is available. Secondly, we use uncorrelated traits by applying a Cholesky transformation to the 34 raw traits. The choice of Cholesky transformation maximises the amount of data used by ordering the traits from the trait with the most complete data to the trait with the least complete data. Each CT trait is the raw trait corrected for all traits before it in this order. Transformed traits are always difficult to interpret but some

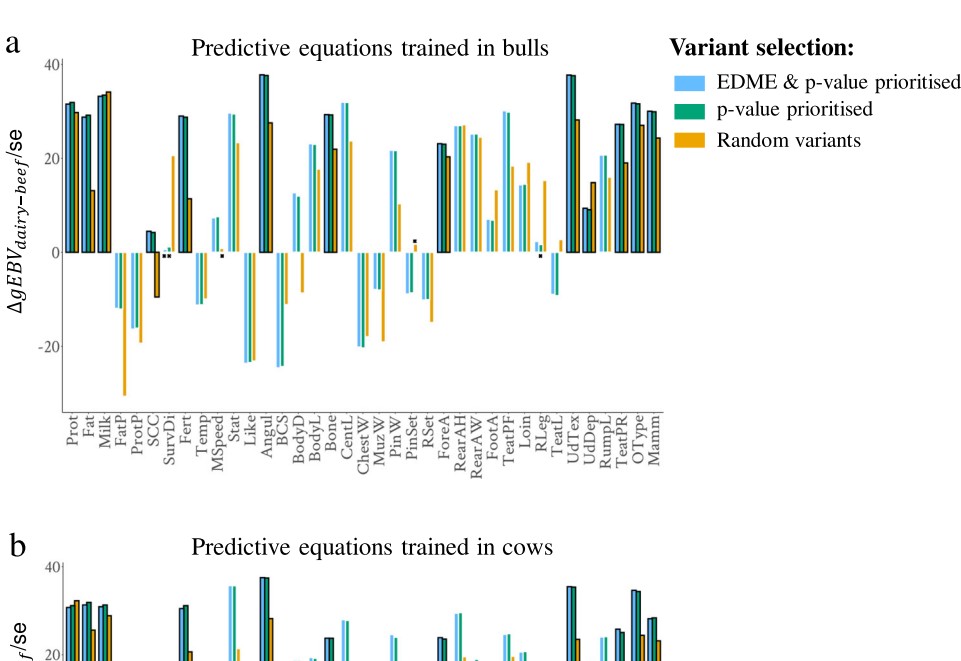

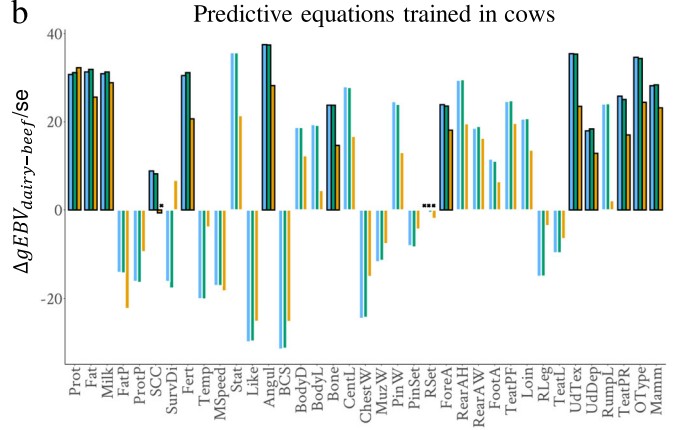

**Fig. 7 Validation of variant selections.** Each bar represents the $t$ value calculated as the mean difference ($\Delta$) of the genomic estimated breeding value (gEBV) between dairy and beef cattle in the 1000-bull database divided by the standard error (se) of that mean difference. One can interpret each bar as a signed $t$ value for the difference of gEBV between dairy and beef cattle and the reference for the mean difference comparison was the beef cattle. The gEBVs were predicted by variants selected using the variant × trait cut-off estimated by the EDME model on top of $p_{wm} < 1e{-}6$ (blue bars), by $p_{wm} < 1e{-}6$ alone based on weighted meta-analysis (green bars), and random variants (yellow bars). The predictive equations were trained in bulls (**a**) and cows (**b**). As labelled by black ×, some comparisons have the value of the mean differences between dairy and beef cattle not significantly different from 0. Bars with thick black borders are the raw traits expected to have larger gEBV in dairy cattle compared to beef cattle. Note that for raw SCC, fertility Fert, Temp, MSpeed and Like, larger gEBV indicates poorer dairy trait performance.

interpretation of the CT traits is still possible. For instance, survival corrected for milk production traits focusses on reasons for death or culling other than milk production. However, when the number of traits considered increases, CT traits ordered late in the list would have reduced interpretability, but this does not affect the estimation of the multi-trait $p$ value, FDRed and number of traits associated with each variant. Where GWAS is already conducted for many potentially correlated traits, it is also possible to decorrelate the variant-trait association matrix as reported by Jordan et al.[7]. However, as the Jordan et al. stated, traits decorrelated using their approach do not necessarily correspond to any specific biological traits of interest[7]. While we do not aim to distinguish between horizontal and vertical pleiotropy, CT traits have a natural interpretation for vertical pleiotropy or Mendelian randomisation where the causal trait precedes targeted traits. However, we have limited power to distinguish cause and effect. Thirdly, we use two separate populations (bulls and cows) to estimate the FDR and the number of true discoveries. This method does not depend on the exact calculation of $p$ values which depends on a statistical model that may not describe the data exactly.

The metric FDRed provides an alternative and powerful approach to controlling GWAS false discoveries. The FDRed estimated from the sign agreement between results from bulls and

cows is smaller than the conventional FDR when variants are chosen that pass a significance test in both sexes. It is possible that variants with small effects but enriched in functional variants can be prioritised by the FDRed (Supplementary Note 1). When variants are selected based on one population only, FDRed is higher than the conventional FDR. This is probably explained by slight differences between the bulls and cows. The bulls come from a slightly different population (e.g., older) and therefore the pattern of LD was different between cows and bulls. Also, the phenotypic values in cows contained larger error variances than bulls[11], therefore, the power for detecting lead variants across 34 traits was different between these two populations, as shown in heritability estimates (Supplementary Data 1), and the correlations between traits were slightly different (Supplementary Fig. 2). These findings suggest that when validating variants using GWAS summary statistics from different populations, the definition of the trait and LD within the populations should be as similar as possible. The genetic correlation between a trait measured in males and females is not always 1.0 and this could create another difference between GWAS results from the two sexes. However, this did not apply in our case because the 'phenotype' of the bulls came from the performance of their daughters. These insights are important for the FDR control and validation of GWAS.

EDME quantifies the number of traits associated with variants in general and with particular variants based on the agreement of variant effect-directions for multiple traits across two populations. This unique analysis provides additional quantities for pleiotropy which fills a gap in our previous methods where only the multi-trait $p$ value was estimated to describe the overall magnitude of variant effects on multiple uncorrelated traits[9]. As mentioned above, Jordan et al.[7] decorrelated a variant-trait association matrix to study horizontal pleiotropy where they quantified the magnitude and number of associated traits for all variants that entered the study. Our analysis focuses on identifying pleiotropic variants for uncorrelated traits in general and uses the multi-trait $p$ value to prioritise a set of credible pleiotropic variants that were analysed by EDME to eliminate false discoveries ($\Pi > 0$) and dissect pleiotropy. The two types of analysis serve different purposes and data availability in two species.

Among variants with $p < 1e-6$ in both sexes we estimate that the average number of traits affected per variant (TE) was 9.6 and this supports the conclusion that pleiotropic effects on uncorrelated traits are widespread. This conclusion is also reached by Jordan et al.[7] using a different analysis in humans. Widespread pleiotropy was also found by analyses of GWAS of correlated traits in humans[6,8]. However, EDME showed that variants varied widely in the number of traits affected from 1 to 22. In general, variants with a large effect on one trait tended to have small effects on many traits (Fig. 4c, d). In additional, EDME assigned a set of pleiotropic variants to each CT trait (trait-related variants, variants_trait) to inform trait biology which is supported by published data[18] (Supplementary Fig. 7).

We discovered that the lack of sharing of variants between CT traits can to some extent inform causal relationships between raw traits, as supported by MR analysis (Fig. 5 and Supplementary Fig. 8 and Supplementary Data 9). However, a more systematic analysis of raw and CT trait relationships aiming at understanding biology between specific trait groups with larger sample size is required in the future.

A key assumption for conducting EDME is that the GWAS results are obtained for the same traits in two independent populations. It is relatively easy to meet this assumption in dairy cattle since matched phenotypes in independent bull and cow populations are common. It is probably also relatively easy to meet the assumption in humans and mice since large GWAS resources are available[2,3,32]. However, it is not clear how easy this assumption can be met in species like beef cattle, chicken, fish, flies and plants. Future research applying EDME to these species will help better understand the usage of the information of variant effect-direction agreement from GWAS between populations in eliminating false discoveries.

Our study identifies pleiotropic loci that provide biological insights beyond cattle, including the *AKTIP-FTO-IRX3-IRX5* loci where the pleiotropic effects on dairy cattle traits were stronger at the non-*FTO* regions (Fig. 6a). Another noticeable loci is the region on cattle chromosome 19 (42.2–42.4M) enriched with keratin genes (*KRT35-14-16*, Fig. 6b). Since *KRT* genes are highly expressed in epithelial systems[31] and it is mammary epithelial cells that secrete milk, mutations in *KRT* genes might be associated with mammary traits and SCC.

In conclusion, we introduce a set of methods for the EDME using GWAS summary statistics to quantify FDR independent of $p$ values. The multi-trait EDME quantifies the pleiotropy where extreme, prevalent and unevenly distributed pleiotropic effect patterns are comprehensively described. EMDE reliably identifies trait-related variants which help to inform the biology behind complex traits. Identified pleiotropic loci including *FTO* and *KRT* update our fundamental knowledge of pleiotropy and gene regulation in large

mammalian species. Prioritised pleiotropic variants are supported by our validation with the 1000-bull Genome data.

## Materials and methods

**Animals, phenotype and genotype data.** No live animals were used in this study. Phenotype data was based on trait deviations for cows and daughter trait deviations for bulls (Supplementary Data 1). Daughter trait deviations were the average trait deviations of a bull's daughters and all phenotypes were pre-corrected for known fixed effects, with processing done by DataGene, Australia for the official release of National bull breeding values (http://www.datagene.com.au/). Only those bulls' phenotypes which were based on records from more than 15 progeny were included. Phenotype data of up to 11,923 bulls included 6569 CRV bulls (https://www.crv4all-international.com/) with phenotypes derived from their Interbull MACE breeding values (https://interbull.org/ib/interbullactivities) degressed on the Australian scale and converted to the scale of the daughter trait deviation. The remaining 5354 bulls and all 32,347 cows were from DataGene. All these animals included Holstein (9739 ♂ / 22,899 ♀), Jersey (2059 ♂ / 6174 ♀), cross breed (0 ♂ / 2850 ♀) and Australian Red dairy breeds (125 ♂ / 424 ♀).

The genotypes for the above-described bulls and cows included a total of 17,669,372 imputed sequence variants with Minimac3[33,34] imputation accuracy $R^2 > 0.4$. Genotypes were imputed using Run6 version of the 1000-bull genome data[18,35] as the reference set, using Eagle[36] to first phase the data and then Minimac3 for imputation. Sequence variants with minor allele frequency > 0.001 in 11,923 bulls and in 32,347 cows were used in the GWAS.

Phenotype decorrelation was conducted using Cholesky transformation, with the formula of $C_n = L^{-1} g_n$ (Eq. 9, published in ref. [9]). $C_n$ was a $K$ (number of traits)×1 vector of Cholesky scores for animal $n$; $L$ was the $K \times K$ lower triangular matrix of the Cholesky factor which satisfied $LL^t = COV$, the $K \times K$ covariance matrix of raw scores after standardisation as z-scores, $g_n$ was a $K \times 1$ vector of traits for animal $n$. Cholesky transformation decorrelated all traits at once. However, as a result, the $K$th Cholesky-transformed trait can be interpreted as the $K$th original trait corrected for the preceding $K-1$ traits and each Cholesky-transformed trait had a variance of close to 1 (Supplementary Data 1). The reason for the choice of these 34 traits and the trait order to conduct Cholesky transformation was to fully utilise the dataset containing the varying number of phenotypic records. A free consideration of the trait order would require all traits to have complete records (e.g., ref. [9]) and this would result in sample size for all traits being determined by the trait with the smallest number of records, i.e., 1439 bulls and 4086 cows. Another condition required to be met for the current Cholesky transformation was that the same descending order of the selected traits matches between the bull and the cow data sets. This allowed that CT traits were the same in bulls and cows such that the GWAS of one trait in two sexes were comparable.

**Single-trait GWAS.** The 34 decorrelated traits were analysed one trait at a time independently in each sex with linear mixed models using GCTA[37]

$$y = \text{mean} + \text{breed} + bx + a + \text{error}, \quad (10)$$

where $y$ = vector of phenotypes for bulls or cows, breed = three breeds for bulls, Holstein, Jersey and Australian Red and four breeds for cows (Holstein, Jersey, Australian Red and MIX); $bx$ = regression coefficient $b$ on variant genotypes $x$; $a$ = polygenic random effects $\sim N(0, G\sigma_g^2)$ where $G$ = genomic relatedness matrix based on all variants. The above GWAS model was also applied to estimate the total genomic variance for heritability calculations for all traits, but without including the "$bx$" term in the model.

**The conventional multi-trait meta-analysis of single-trait GWAS.** The multi-trait $\chi^2$ statistic for the $i$th variant was calculated based on its signed $t$ values generated from each single-trait GWAS: $\chi^2 = t t_i V^{-1} t_i$ (Eq. 11, published in ref. [1]). $t_i$ was a $K$ (number of traits = 34) × 1 vector of the signed $t$ values of variant$_i$ effects, i.e., $b/se$, for the $K$ traits; $t_i'$ was a transpose of vector $t_i (1 \times K)$; and $V^{-1}$ was an inverse of the $K \times K$ correlation matrix where the correlation was calculated over the all estimated variant effects (signed $t$ values) of all trait pairs. The $\chi^2$ value of each variant was examined for significance based on a $\chi^2$ distribution with $k$ degrees of freedom to test against the null hypothesis that the variant had no significant effects on any one of the $K$th traits. The conventional FDR of $\chi^2$ tests were calculated following[1].

To fully utilise the GWAS summary data obtained in bulls and cows separately, a weighted multi-trait meta-analysis of the signed $t$ values of bull GWAS and cow GWAS results was performed using the formula

$$t_w = \frac{b_w}{se_w} = \frac{\frac{b_{bull}}{se_{bull}^2} + \frac{b_{cow}}{se_{cow}^2}}{\frac{1}{se_{bull}^2} + \frac{1}{se_{cow}^2}} \bigg/ \sqrt{\frac{1}{\frac{1}{se_{bull}^2} + \frac{1}{se_{cow}^2}}}, \quad (11)$$

(published in ref. [12]). Where $t_w$ was the weighted $t$ value for bulls and cows accounting for the phenotypic error differences between bulls and cows[11], calculated as the quotient of the weighted variant effects $B_w$ and the weighted effect

error $se_w$; $b_{bull}$ and $se_{bull}$ were the variant effects and errors obtained from single-trait GWAS in bulls and $b_{cow}$ and $se_{cow}$ were the variant effects and errors from cow GWAS. Once the $t_w$ of each trait was obtained, the multi-trait meta-analysis was performed using Eq. 11. This weighted meta-analysis approach was also used to analyse the GWAS summary statistics data of the 2017 study of cattle pleiotropy[9], where over 630,003 SNPs from the high-density SNP chip panel were used for GWAS with 25 traits in bulls and cows.

**The sliding-window variant clustering**. The use of sequence variants led to a large number of variants in high LD. Consequently, a single QTL might be tagged by many sequence variants. To give an indication of the differentiation of pleiotropic QTL from sequence variants, a sliding-window (5 Mb) variant clustering was conducted. All calculations regarding this clustering were done separately in each sex. Firstly, variants were ranked by their multi-trait $p$ values adjusted (divided) by their functional-and-evolutionary trait heritability (FAETH) score[13]. Then, variants ranked within the top 10% within each sex were selected for LD pruning, using Plink 1.9[38] with $r^2 > 0.95$ within 5 Mb windows. For pruned variants, in each 5 Mb window, with the overlapping size of 2.5 Mb, we calculated $\rho_{ij} = r_{cor}(t_{variant_i}, t_{variant_j}) \times r_{LD}(variant_i, variant_j)$, where $r_{cor}(t_{variant_i}, t_{variant_j})$ was the correlation between two vectors of $t$ values (beta/s.e. from GWAS described above) of 34 CT traits for $variant_i$ and $variant_j$; $r_{LD}(variant_i, variant_j)$ was the LD ($r$) measured as the correlation between the genotypes of $variant_i$ and $variant_j$. These $\rho$ values were clustered using graph based Random Walks determining densely connected subgraphs (clusters)[39] implemented in igraph[40]. On average there were 12 variants per cluster in bulls and cows. The clustering of the variants aimed at finding groups of variants tagging the same QTL. Normally, variants with strong LD were considered as tagging the same QTL. However, variants tagging one QTL should also show similar effect patterns across multiple traits (correlated GWAS $t$ values). Therefore, by adding the correlation of effect $t$ value, i.e., $r_{cor}(t_{variant_i}, t_{variant_j})$ on top of the $r_{LD}(variant_i, variant_j)$, we expect the metric $\rho_{ij}$ to better group variants tagging the same QTL than using $r_{cor}$ or $r_{LD}$ alone. After clustering, up to 50 clusters were retrieved within each window. To verify if identified clusters can represent local QTL, the top 100 variants from clusters based on their FAETH adjusted multi-trait $p$ value were selected. These 100 top variants were used to conduct a conditional analysis of GWAS[37] of 34 traits in each sex. A meta-analysis of the conditional GWAS results used Eq. 11.

Meta-analysis of the conditional GWAS results showed that pleiotropic effects dramatically dropped, but only for those variants from the clusters with which the top 100 variants were selected (Supplementary Fig. 5). However, the pleiotropic effects of those variants from the clusters not used to select the top 100 variants did not show significant drops (Supplementary Fig. 5). These results support the hypothesis that, in general, pleiotropic QTL can be differentiated from sequence variants, but no doubt there were exceptions to this conclusion.

**The single-trait EDME model**. The R code of EDME can be downloaded from https://figshare.com/s/42017bcae071d24639f4 and an online tutorial for conducting EDME of GWAS is available at: https://ruidongxiang.com/effect-direction-meta-analysis-edme-of-gwas/. In theory, there should be three categories of variants associated with a given trait: (1) variants with 'true negative' effects on the trait, i.e., $T_-$ variants that truly decrease the trait value, (2) variants with 'true positive' effects on the trait, i.e., $T_+$ variants that truly increase the trait value and (3) variants with 'true zero' effects on the trait, i.e., $T_0$ variants that had no effects on the trait. In practice, a GWAS conducted for the same trait in two different populations would have four categories of variant effect directions: (1) the '$--$' where the effect was consistently negative for the trait in two populations, e.g., bulls and cows in this study; (2) the '$++$' where the effect was consistently positive for the trait in bulls and cows; (3) the '$+-$' where the effect was positive for the trait in bulls but negative in cows; and (4) the '$-+$' where the effect was negative for the trait in bulls but positive in cows.

If there were $T_0$ variants which should have no TE, they were expected to be randomly distributed across the four observed categories ('$++$', '$--$', '$+-$' and '$-+$') of variants. This meant that each one of the four categories of the '$++$', '$--$', '$+-$' and '$-+$' variants contain an equal amount of the $T_0$ variants (i.e., one-fourth). Therefore, the total number of '$+-$' and '$-+$' variants observed was equal to one-half of the total number of $T_0$ variants expected. This then allowed the deduction of model Eq. 1, as described in the section 'Results', to calculate the number of $T_0$. Once the number of $T_0$ was known, the total number of $T_-$ and $T_+$ variants can be determined by equating the expectation with the observed number of '$++$' and '$--$' variants ($n(B)$) as described in the first and the third quadrant of the right panel of Fig. 2a.

$$n(B) = T_+ + \frac{T_0}{4} + T_- + \frac{T_0}{4} = T_+ + T_- + \frac{T_0}{2},$$

then

$$n(B) - \frac{T_0}{2} = T_- + T_+ = T_r. \qquad (2)$$

Once the number of $T_0$ and $T_r$ were determined, the false discovery rate by effect direction (FDRed) and true discovery rate by effect direction (TDRed) can be calculated based on the total number of variants entered the analysis in the two populations ($n(A) + n(B)$) as described in Eqs. 3 and 4 (see Results section).

**The EDME model for multi-trait analysis**. For equation $\Pi_i = t_{i1} \cdot t_{i2}^T$ (5), the $\Pi$ value was calculated as the dot product of the vectors of $t$ values of bulls and cows with the same length of $K$, where $K$ is the total number of traits. The idea was that the effect direction across multiple traits, i.e., the sign of the $\Pi$ value, was not driven by the small trait effects which were more likely to arise from noise. Instead, the sign of the $\Pi$ value was driven by relatively large trait effects which were likely to be robust. Then, the number of variants with negative $\Pi$ values were used to calculate the multi-trait FDRed in Eq. 6.

For one variant analysed in K traits across the two populations, the single-trait Eq. 2 can be extended as

$$n(B)_{MTR} = \left( P_{T_r} + \frac{P_{T_0}}{2} \right) \times K, \qquad (13)$$

where $n(B)_{MTR}$ was the multi-trait version of $n(B)$ that was equivalent to $\sum_1^K n(B)_{MTR}$ (in Eq. 7), where $n(B)_{MTR} = 1$ if the variant has the same effect direction and $n(B)_{MTR} = 0$ if the variant has different effect directions between the two populations. $P_{T_0}$ was the proportion of the total number of variants with zero effects ($T_0$) and $P_{T_r}$ was the proportion of the total number of variants with TE ($T_r$). As $P_{T_r} + P_{T_0} = 1$; Eq. 13 was solvable and can be re-arranged as

$$P_{T_r} = \frac{2 \times n(B)_{MTR}}{K} - 1 = \frac{2 \times \sum_1^K n(B)_{MTR}}{K} - 1 = \text{multi-trait TDRed},$$

then

$$\text{multi-trait TDRed} \times K = 2 \times \sum_1^K n(B)_{MTR} - K = \text{estimated number of TE} = Eq. 7,$$

where TE is the True Effect of a variant on traits. Note that if the number of traits considered was one ($K = 1$), the multi-trait Eq. 13 was then identical to single-trait Eq. 2 at the probability level. $\text{TDRed}_{MTR}$ was the multi-trait version of TDRed for Eq. 4.

For Eq. 8, when many variants were selected, the estimated number of TE averaged across a group of variants, i.e., $\overline{TE}$, became stable. Thus, the number of variants with positive $\Pi$ values $n(\text{variant})$ multiplied by the number of $\overline{TE}$ will inform the TC of trait by variant, $TC_{Tr \times V}$. For single-trait GWAS of K traits, each one of the total X number (e.g., 17 mil) of variants would have K number (e.g., 34) of $t$ values. Then, for a total X number of variants by K number of traits, a $X \times K$ length (e.g., 17 mil × 34) vector of $t$ values can be obtained. If $TC_{Tr \times V} = n(\text{variant}) \times \overline{TE} = 80,000 \times 10 = 800,000$, then the top 800,000 combinations of trait by variant ranked by their $t$ value (absolute value) in the $X \times K$ vector would be prioritised by Eq. 8. The unique variants with $\Pi > 0$ in these combinations of trait by variant were the prioritised pleiotropic variants.

Once a prioritised set of $X \times K$ TC of trait by variant was achieved, two additional pieces of information can be estimated. One was the count of the traits assigned to each variant in the $X \times K$ vector and this was the observed TE for each variant. The other was the count of the variants with positive $\Pi$ assigned to each trait in the $X \times K$ vector and this was the identification of the pleiotropic variants related to each trait, $\text{variants}_{trait}$. See the online tutorial for using EDME to quantify pleiotropy and to do other related analyses at https://ruidongxiang.com/effect-direction-meta-analysis-edme-of-gwas/.

**Sharing of trait-related variants ($\text{variants}_{trait}$)**. The number of pleiotropic $\text{variants}_{trait}$ in overlaps between 34 CT trait sets were counted. The significance of overlaps were compared with the expected number using the hypergeometric test ($p$) using the variant-overlap matrices generated by GeneOverlap[41] in R. This analysis required four types of counts: the size of overlap between set A (e.g., variants assigned to trait 1) and set B (e.g., variants assigned to trait 2), the size of set A, the size of set B and the size of background (92,537 prioritised by EDME). The number of variants in the overlap and in the union of pleiotropic $\text{variants}_{trait}$ between traits was obtained. The number of overlapping dividing the number of the union was used to calculate the sharing index for each trait pair. For each trait, its sharing indices with the other 33 traits (excluding self-pair) were averaged and the averaged sharing index was then used to rank each trait.

**Conventional hierarchical cluster analysis**. This clustering was based on the correlation matrices for weighted $t$ values (Eq. 11) of variants between bulls and cows. The correlation matrices were used to perform hierarchical clustering and were shown as heatmaps using 'complexHeatmap'[42] in R v3.4.1.

**Variant annotation**. Variants were annotated using the Variant Effect Prediction of Ensembl[22] and NGS-SNP[43] based on the Run 6 version of the 1000-bull genome data.

**Mendelian randomisation**. This analysis was conducted using GSMR[20]. Summary statistics of GWAS of raw traits (daughter trait deviations, not Cholesky transformed) for those 93,513 variants prioritised by $p$ value and multi-trait EDME were used as variant input for GSMR. Settings of MR followed the default options of GSMR, including the threshold of HEIDI outlier test being $p < 0.01$. Both forward and reverse analysis were conducted. GSMR generated MR $p$ values were corrected for multi-testing by the Bonferroni method. A putative causal relationship was defined as the forward MR was significant and the reverse MR was not significant at Bonferroni corrected $p < 0.05$ level.

**Further analysis of variants$_{trait}$**. For each set of variants$_{trait}$ identified by the multi-trait analysis, their single-trait GWAS from bulls and cows were mapped and were analysed for single-trait FDRed using Eq. 3. Totally, 164 previously prioritised variants for cattle stature were retrieved from ref. [18] and their enrichment in each set of variants$_{trait}$ was tested using the hypergeometric test. The $p$ value of enrichment was adjusted for multi-testing.

**Joint analysis**. The joint analysis was conducted using GCTA-COJO[37] function. The model for the joint analysis was the same as Eq. 10 expect that selected variants were fitted together. The joint analysis was carried out for each one of the 37 raw traits in bulls and cows. The weighted $t$ value was re-calculated based on the beta ('bJ') and se ('seJ') generated from the joint analysis for each trait using Eq. 11. Then the weighted $t$ values were presented in Fig. 6a, b.

**Validation in the 1000-bull genome data**. The 1000-bull genome database (Run 6[18,35]) contained the whole genome sequence of cattle independent of the Australian data with no phenotype but with breed information. To perform a validation test without phenotype in the validation population, we used the genotypes of selected variants and 37 raw traits (Supplementary Data 11) of the Australian dairy cattle to generate genomic prediction equations separately in the bull and cow populations. Then we used these prediction equations to predict the phenotype of each 1000-bull individual (i.e., generating two sets of genomic estimated breeding values, gEBV, for every raw trait). If a variant selection was informative, the average gEBV of the 1000-bull genome dairy cattle (e.g., Holstein breed) was expected to be better than the average gEBV of 1000-bull genome beef cattle (e.g., Angus).

The genomic best linear unbiased prediction (gBLUP) implemented in MTG2[44] was used to train genomic prediction equations in each of the Australian bull and cow data sets. The genomic prediction equations were generated for three different variant sets: (1) prioritised by EDME and $p$ value, (2) prioritised by $p$ value alone and (3) a random set of variants with the matching variant number. These 3 sets of genomic prediction equations were generated for each of the 37 original traits (raw daughter trait deviations, not Cholesky transformed) in both the bull and cow data sets. The summary statistics of these 37 raw traits are given in Supplementary Data 11. The gBLUP model was the same as Eq. 10 except that no individual variants were fitted (i.e., removing "bx"). This estimated the total genetic value of Australian bulls and cows and allowed back-solving for the variant effect solutions in the Australian data (-sbv option). Then, the variant effect solutions were then used to calculate gEBV of the dairy and beef breed individuals in the 1000-bull genome (Run 6, $N = 2334$).

The dairy cattle in the 1000-bull genome included Holstein ($N = 567$), Jersey ($N = 66$), Brown Swiss ($N = 148$), Finnish Ayrshire ($N = 25$), Normandy ($N = 44$), Norwegian Red ($N = 24$), Swedish Red ($N = 16$). The 1000-bull beef cattle included Angus ($N = 272$), Beef Booster ($N = 29$), Belgian Blue ($N = 16$), Blonde d'Aquitaine ($N = 26$), Braunvieh Beef ($N = 4$), Charolais ($N = 128$), Hereford ($N = 75$), Limousin ($N = 82$), Maine Anjou ($N = 5$) and Simmental ($N = 225$). The mean of gEBV of the 1000-bull dairy and beef cattle were compared using a $t$ test. For the $t$ test of each trait between dairy and beef cattle, a $t$ value was calculated as the mean difference (with reference level set to beef cattle) divided by the standard error and this $t$ value was presented in Fig. 7.

**Reporting summary**. Further information on research design is available in the Nature Research Reporting Summary linked to this article.

## Data availability

The DNA sequence data as part of the 1000-bull genome project[18,35,45] included PRJNA431934, PRJNA238491, PRJDB2660, PRJEB18113, PRJEB1829, PRJEB27309, PRJEB28191, PRJEB9343, PRJNA210519, PRJNA210521, PRJNA210523, PRJNA279385, PRJNA294709, PRJNA316122, PRJNA474946, PRJNA477833, PRJNA494431, PRJDA48395, PRJNA431934 and PRJNA238491. The variant FAETH score is available at https://doi.org/10.26188/5c5617c01383b. All other data supporting results and figures are shown in the Supplementary Data and Figures.

## Code availability

The GWAS was conducted using public software Plink[38] and GCTA[37]. Variant annotation used public Variant Effect Prediction of Ensembl[22] and NGS-SNP[43]. The code of EDME software implemented in R with sample datasets and demo workflow can be downloaded from https://figshare.com/s/42017bcae071d24639f4. An online tutorial showing how to use the code and sample data to conduct EDME of GWAS is available at https://ruidongxiang.com/effect-direction-meta-analysis-edme-of-gwas/.

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

## Acknowledgements

Australian Research Council's Discovery Projects (DP160101056) supported R.X. and M.E.G. DairyBio (a joint venture project between Agriculture Victoria and Dairy Australia) funded computing resources used in the analysis. I.B. was supported by the Center for Genomic Selection in Animals and Plants (GenSAP) funded by Innovation Fund Denmark (grant 0603-00519B). No funding bodies participated in the design of the study nor analysis, or interpretation of data nor in writing the manuscript. DataGene and CRV (www.crv4all-international.com/) provided access to data used in this study. We thank Gert Nieuwhof, Kon Konstantinov and Timothy P. Hancock (DataGene) and Chris Schrooten (CRV) for preparation and provision of data. We thank partners from the 1000-bull genome project for the data access. We thank Dr. Mekonnen Haile-Mariam for deriving the deregressed phenotypes from international MACE and Dr. Sunduimijid Bolormaa for sequence variant data imputation.

## Author contributions

M.E.G. and R.X. conceived the study. R.X. and I.B. implemented the analysis. I.M.M. and H.D.D. provided data and assisted with study design. R.X., I.B. and M.E.G. analysed the data. R.X. developed the EDME software. R.X. and M.E.G. wrote the paper. R.X., M.E.G., I.B., I.M.M. and H.D.D revised the paper. All authors read and approved the final paper.

## Competing interests

The authors declare no conflict of interest.
