## [Peer Review File · Communications Biology]

Reviewers' comments:

Reviewer #1 (Remarks to the Author):

Review of Xiang et al. Effect direction meta-analysis of GWAS identifies extreme, prevalent and shared pleiotropy in a large mammalian model

Authors present a method of using meta analysis of multiple independent samples to identify reliable pleiotropy across uncorrelated traits. They use an alternate method to assess FDR and incorporate information about the direction of effects to assess true variant effects. The use of uncorrelated traits is important and helps to uncover true pleiotropy, as opposed to other complex situations in which one gene can appear to affect multiple processes through downstream effects. The authors are also careful to distinguish pleiotropy and LD using a novel variant clustering method. The independent validation method was clever and demonstrates both the validity and the practicality of the approach.

The paper clearly explains and motivates the method. The data sets used seem ideal for this kind of analysis.

I think the results and methods are of interest to a broad audience.

Comments:

I don't understand how the Cholesky transformed traits can still be called by raw trait names. Aren't these composite traits? It would be nice to see a more fleshed out explanation of this, perhaps with a figure showing the correlation between traits and how the original traits relate to the Cholesky transformed traits.

Why does ρ_{ij} use both LD and the correlation between the t statistics? Can you discuss in the paper what the inclusion of the t statistic correlation adds?

What considerations should be made when applying this type of analysis to populations of other organisms, for example, humans, mice, or *Drosophila*? Are there aspects of the cattle population that make it particularly suited to this type of analysis? Should changes be made to the pipeline when moving to a different type of population? Or are there particular assumptions that should be checked before applying this analysis to a different type of population?

The text should be proof-read carefully for typos and to make sure type face, capitalization, and tense conventions are followed consistently.

Minor suggestions:

It would help readability to make the axis labels in the figures a bit larger

There is no x-axis label in Figure 3A

Reviewer #2 (Remarks to the Author):

In the literature there are already studies suggesting the association between physical activity and the reduction of the risk of breast and colorectal cancers. The authors investigated their causal link by a Mendelian randomization approach.

The study is well conducted and presented. I suggested some minor comments.

1. Given the small number of GWA-significant SNPs used as instruments, they could try to investigate the presence of pleiotropy from a biological point of view.

2. They could provide some plots to investigate the directional pleiotropy.
3. I suggest to revise the notation, to be consistent.

Reviewer #3 (Remarks to the Author):

This manuscript describes a novel method for performing effect direction meta-analysis on GWAS results. The method is clever and well worked out. However, after reading the manuscript, I remain unclear about the purpose of the method and its benefits over other existing methods, as well as about certain methodological choices the authors make and the novelty of their findings. The manuscript would benefit greatly from clearer discussion of these issues.

I have three major issues I would like to see discussed:

1. I do not have a clear sense of the motivation for developing this method, and particularly of what advantages it has over existing methods. The authors specifically mention two existing methods in the manuscript. The first is Turley et al.'s MTAG method, which the authors claim cannot be used in cattle due to its requirement for precomputed LD scores. I primarily work in humans, so I may not be familiar with these limitations, but it is not obvious to me which assumptions of LD score regression do not apply in cattle, or why the LDSC software package cannot be used to compute LD scores from a cattle population sequencing dataset just as easily as from a human population sequencing dataset. A quick search on Google Scholar shows multiple published papers that have used LD score regression in cattle. The second method the authors mention is the same analysis they perform using Cholesky-decorrelated traits, but using a more conventional FDR calculation based on p-values. It is not clear to me that the conventional p-values can't be used to do the exact same analysis of pleiotropy performed here. It appears that many of the same authors did a version of that analysis in Xiang et al. 2017, though I may be mistaken as I did not read that paper in detail for this review. At any rate, there is very little direct comparison of results between the conventional FDR method and the EDME method, which makes it very difficult to evaluate how much benefit the new method gives over the old. In fact, in the one figure that does show a direct comparison, Figure 7, the two methods show nearly identical results. The manuscript would benefit greatly from more discussion of both of these existing methods (MTAG and conventional FDRs), and either including more detail about why the existing methods are unsuitable for the analyses done in this manuscript, or showing the results those methods would produce for figures 1,3,4,5,6, and Table 1, so that the advantage the EDME method provides can be quantified.

2. I find the Cholesky decorrelation method very confusing, and practically no explanation of it is provided in this manuscript. I understand in general terms that the idea is to guarantee that the traits being measured have no overall correlation with each other, but I do would like to see some discussion of why this is important to the analysis. Additionally, I do not understand why the traits are ordered sequentially, rather than being compiled into a single matrix and decorrelated all at once, or why ordering by missingness is appropriate. This is an extremely important methodological detail, because it seems to me that the entire interpretation of the analysis is dependent on the ordering that is chosen. For example, the authors point out in the section "Trait relationships informed by Cholesky decorrelation" that a variant that affects protein yield and survival by corresponding amounts would be treated as having an effect on protein yield and no effect on survival. As a result, if that same variant had an effect on a trait later in the list, such as center ligament (trait 11), that would be counted as a shared variant between protein yield and center ligament, but not between survival and center ligament. How much do the results and interpretation of the analysis change if the order of traits is permuted? Wouldn't it make more sense to use some kind of Mendelian randomization or mediation analysis to determine which traits to assign effects to, rather than assuming that the trait

with the least missing data is the true causal trait?

3. The authors argue that the ability to study pleiotropic effects of variants is novel, and go so far as to state in the Discussion that the kind of widespread pleiotropy they observe here has never before been reported in mammals. As I mentioned above, I primarily work in humans, but in the field of human genetics I am aware of at least three papers published in the last year that report widespread pleiotropy, some of them using similar methods to this manuscript: Verbanck et al. 2018 (<https://www.nature.com/articles/s41588-018-0099-7>), Watanabe et al. 2019 (<https://www.nature.com/articles/s41588-019-0481-0>), and Jordan et al. 2019 (<https://genomebiology.biomedcentral.com/articles/10.1186/s13059-019-1844-7>). (In the interest of full disclosure, I am an author on one of these papers.) A great deal has been written about the presence of pleiotropy in GWAS traits, and much of it in the last few years has been highlighting the pervasiveness of pleiotropy and what that implies about the genetic architecture of traits. Using a multi-trait GWAS meta analysis to identify widespread pleiotropy in a mammalian species is not novel in itself. That is not to say that this manuscript does not contain novel results with interesting implications for the field, but more discussion of the context and the current state of the field is required.

In addition to these three major issues, there are a large number of typos, grammatical errors, and poor formatting that make the manuscript fairly difficult to follow in places. I am not going to go through the manuscript in detail to find all these errors, because a peer reviewer should not have to be a copy editor, but please proofread carefully and make sure all the figures and tables have appropriate legends, labels, and captions.

Letter of response to referees

MS# COMMSBIO-19-1181-T, 'Effect direction meta-analysis of GWAS identifies extreme, prevalent and shared pleiotropy in a large mammalian model'

We thank the reviewers for their effort and time in providing feedback on the manuscript. We have revised our manuscript accordingly. Please find the detailed point-by-point response to the reviewers in the following text. The author's response is in blue text.

Reviewer #1 (Remarks to the Author):

Review of Xiang et al. Effect direction meta-analysis of GWAS identifies extreme, prevalent and shared pleiotropy in a large mammalian model

Authors present a method of using meta analysis of multiple independent samples to identify reliable pleiotropy across uncorrelated traits. They use an alternate method to assess FDR and incorporate information about the direction of effects to assess true variant effects. The use of uncorrelated traits is important and helps to uncover true pleiotropy, as opposed to other complex situations in which one gene can appear to affect multiple processes through downstream effects. The authors are also careful to distinguish pleiotropy and LD using a novel variant clustering method. The independent validation method was clever and demonstrates both the validity and the practicality of the approach.

The paper clearly explains and motivates the method. The data sets used seem ideal for this kind of analysis.

I think the results and methods are of interest to a broad audience.

Author repose: We appreciate the positive comments from the reviewer.

Comments:

I don't understand how the Cholesky transformed traits can still be called by raw trait names.

Aren't these composite traits? It would be nice to see a more fleshed out explanation of this, perhaps with a figure showing the correlation between traits and how the original traits relate to the Cholesky transformed traits.

Author response: We apologize for this confusion between the Cholesky transformed traits and the original traits. To clarify this, throughout the manuscript, we have added the label 'CT' to traits when describing results specifically related to Cholesky-transformed traits. When describing results related to original traits that were not Cholesky-transformed, a label 'raw' was used. We have provided descriptions of this labeling upfront in the second paragraph of Results '*Single-trait GWAS and conventional multi-trait meta-analysis in bull and cow populations*' where Cholesky transformation is explained. In this paragraph, we also provided extra explanations of the Cholesky transformation.

As requested, and a new Supplementary Figure S1 is added to further explain the links between the original traits and the transformed traits. A new paragraph (the current 1st paragraph in Discussion) has been added to the Discussion to further explain the motivation and conditions for the Cholesky transformation. To summarise, the Cholesky transformation decorrelated all traits at once. It can be described by saying that the Kth Cholesky transformed trait can be interpreted as the Kth original trait corrected for the preceding K-1 traits. By ordering the traits in a fashion that the traits with most complete data come first, we make maximal use of the data. This is because all animals recorded for the Kth trait can be used to calculate the Kth Cholesky transformed trait, as this calculation only requires data from the preceding K-1 traits and animals with the Kth trait recorded have data on the K-1 preceding traits. If we had complete data for 34 traits matched in bulls and cows (e.g., our 2017 study) then we could freely consider the order of the Cholesky-transformation. However, this was not the case and having 34 traits with complete records allowing for free consideration of the trait order would result in the sample size for all traits being determined by the trait with the smallest number of records, i.e., 1439 bulls and 4086 cows (instead of up to 11923 bulls and up to 32347 cows).

Why does rho_ij use both LD and the correlation between the t statistics? Can you discuss in the paper what the inclusion of the t statistic correlation adds?

Author response: we have added the following text to explain the purpose of using rho_ij in the 7th paragraph of the Materials and methods (section 'The sliding-window variant

clustering.’): ‘The clustering of the variants aimed at finding groups of variants tagging the same QTL. Normally, variants with strong LD were considered as tagging the same QTL. However, variants tagging one QTL should also show similar effect patterns across multiple traits (correlated GWAS t values). Therefore, by adding the correlation of effect t value, i.e., $r_{cor}(t_{variant_i}, t_{variant_j})$ on top of the $r_{LD}(variant_i, variant_j)$, we expect the metric ρ_{ij} to better group variants tagging the same QTL than using r_{cor} or r_{LD} alone.’

What considerations should be made when applying this type of analysis to populations of other organisms, for example, humans, mice, or Drosophila? Are there aspects of the cattle population that make it particularly suited to this type of analysis? Should changes be made to the pipeline when moving to a different type of population? Or are there particular assumptions that should be checked before applying this analysis to a different type of population?

Author response: we have added the following text as the new 8th paragraph in Discussion: ‘A key assumption for conducting EDME is that the GWAS results are obtained for the same traits in two independent populations. It is relatively easy to meet this assumption in dairy cattle since matched phenotypes in independent bull and cow populations are common. It is probably also relatively easy to meet the assumption in humans and mice since large GWAS resources are available [1-3]. However, it is not clear how easy this assumption can be met in species like beef cattle, chicken, fish, flies and plants. Future research applying EDME to these species will help better understand the usage of the information of variant effect-direction agreement from GWAS between populations in eliminating false discoveries.’

The text should be proof-read carefully for typos and to make sure type face, capitalization, and tense conventions are followed consistently.

Author response: We apologize for these errors. We have carefully proof-read the manuscript to eliminate these errors. We have re-generated figures to make sure they all have a consistent typeface.

Minor suggestions:

It would help readability to make the axis labels in the figures a bit larger

Author response: in the revised manuscript, we have increased the size of the axis labels in most of the figures.

There is no x-axis label in Figure 3A

Author response: We apologize for this error and we have added the x-axis label ($-\log_{10}(\text{conventional FDR})$) into Figure 3A.

Reviewer #2 (Remarks to the Author):

In the literature there are already studies suggesting the association between physical activity and the reduction of the risk of breast and colorectal cancers. The authors investigated their causal link by a Mendelian randomization approach.

The study is well conducted and presented. I suggested some minor comments.

Author response: we thank the reviewer for the positive comments and we have revised the manuscript accordingly.

1. Given the small number of GWA-significant SNPs used as instruments, they could try to investigate the presence of pleiotropy from a biological point of view.

Author response: we agree with the reviewer that the Mendelian randomisation analysis lacked power and this limitation along with others have been acknowledged in the revised manuscript (7th paragraph of Discussion). We also agree with the reviewer that the mendelian randomisation and pleiotropy analysis led by biological hypotheses would be more useful. However, our study has focused on quantifying false discoveries and identifying pleiotropic variants associated with uncorrelated traits using the agreement of variant effect directions. We believe that there have been enough results related to these primary foci presented in the manuscript. Therefore, we wish to pursue the additional work more systematically in the future as mentioned at the end of the 7th paragraph of the Discussion: ‘A more systematic analysis of raw and CT trait relationships aiming at understanding biology between specific trait groups with larger sample size is required in the future.’

2. They could provide some plots to investigate the directional pleiotropy.

Author response: we appreciate the suggestion from the reviewer. As requested we have added a new Supplementary Figure S8 with additional plots, including funnel plots, from the MR analysis of protein and survival to the manuscript. In the text, a statement ‘Additional plots of the MR analysis indicated the existence of directional pleiotropy [4, 5] in the MR of survival $\xrightarrow{\text{cause}}$ protein, but not in the MR of protein $\xrightarrow{\text{cause}}$ survival (Supplementary Figure S8).’ has been added to the second paragraph of the section ‘*Raw trait relationships informed by Cholesky decorrelation*’ in the Results.

3. I suggest to revise the notation, to be consistent.

Author response: we apologise for these inconsistencies. We have carefully checked the notations and revised them to be consistent.

=====

Reviewer #3 (Remarks to the Author):

This manuscript describes a novel method for performing effect direction meta-analysis on GWAS results. The method is clever and well worked out. However, after reading the manuscript, I remain unclear about the purpose of the method and its benefits over other existing methods, as well as about certain methodological choices the authors make and the novelty of their findings. The manuscript would benefit greatly from clearer discussion of these issues.

Author response: we appreciate the positive comments of the reviewer and apologise for not being updated with the literature in humans which led to the unclear motivation. We have extensively revised the Introduction and Discussion to include up-to-date references as suggested by the reviewer and clarify our aims. In the significantly restructured Discussion we also discuss these references with our analysis. In short, our analysis has several distinctions from many pleiotropy studies including the references pointed out by the reviewer: 1) our study starts with the GWAS of each cattle trait instead of using pre-existed GWAS summary statistics (e.g., UK biobank); 2) our study focuses on pleiotropy among traits decorrelated by Cholesky-transformation and 3) we aim to use the agreement of variant

effect-direction to quantify FDR independent of p values and; 4) to identify pleiotropic variants in general, instead of to distinguish different types of pleiotropy type (e.g., vertical/horizontal). We specify these points in the following response to the comments of the reviewer and in revisions in the updated manuscript.

I have three major issues I would like to see discussed:

1. I do not have a clear sense of the motivation for developing this method, and particularly of what advantages it has over existing methods. The authors specifically mention two existing methods in the manuscript. The first is Turley et al.'s MTAG method, which the authors claim cannot be used in cattle due to its requirement for precomputed LD scores. I primarily work in humans, so I may not be familiar with these limitations, but it is not obvious to me which assumptions of LD score regression do not apply in cattle, or why the LDSC software package cannot be used to compute LD scores from a cattle population sequencing dataset just as easily as from a human population sequencing dataset. A quick search on Google Scholar shows multiple published papers that have used LD score regression in cattle. The second method the authors mention is the same analysis they perform using Cholesky-decorrelated traits, but using a more conventional FDR calculation based on p-values. It is not clear to me that the conventional p-values can't be used to do the exact same analysis of pleiotropy performed here. It appears that many of the same authors did a version of that analysis in Xiang et al. 2017, though I may be mistaken as I did not read that paper in detail for this review. At any rate, there is very little direct comparison of results between the conventional FDR method and the EDME method, which makes it very difficult to evaluate how much benefit the new method gives over the old. In fact, in the one figure that does show a direct comparison, Figure 7, the two methods show nearly identical results. The manuscript would benefit greatly from more discussion of both of these existing methods (MTAG and conventional FDRs), and either including more detail about why the existing methods are unsuitable for the analyses done in this manuscript, or showing the results those methods would produce for figures 1,3,4,5,6, and Table 1, so that the advantage the EDME method provides can be quantified.

Author response: We have extensively revised the introduction to include more references from humans and to clarify the aims. As stated in the new 2nd and 3rd paragraphs of the Introduction, the aims are (1) to use the EDME model to calculate an FDR (FDRed) which does not rely on p values and (2) the use of the EDME model to describe the extent of

pleiotropy. FDRed is an advantage over conventional FDR because p-values are model dependent and the data does not always conform to the statistical model used to analyse it. More importantly, the EDME model can be extended to estimate the number of traits affected by a variant which we believe is novel. Conventional FDR does not provide this information.

LDSC regression can be used in cattle and we apologise if we gave a different impression. For instance, it could be used to estimate the genetic correlation between trait but that is not our aim. Nor is most of the paper concerned with the distinction between horizontal or vertical pleiotropy although we do briefly consider Mendelian randomization. MTAG based on LDSC regression is designed to estimate the effect of a variant on a set of correlated traits. As stated by the authors, if the traits are uncorrelated, MTAG is equivalent to single-trait analysis and indicates little about pleiotropy. However, the aim of our paper is to evaluate the extent of pleiotropy among uncorrelated traits. Therefore, we have revised the 4th paragraph in the Introduction where we removed those sentences related to LDSC regression and focused on our aim that ‘Naturally, traits that are genetically correlated must share part of their causal variants, however, if traits are uncorrelated, can we still detect widespread pleiotropy?’. To emphasise our aim, a sentence ‘After GWAS, we focus on using the information of the consistency of variant effect directions to identify pleiotropic variants associated with uncorrelated traits, instead of distinguishing different types of pleiotropy’ has been added to the last paragraph of the Introduction.

We have extensively revised the Discussion to clarify the feature of our analysis. EDME has two major features: 1) to calculate the false discovery rate by effect direction (FDRed) 2) based on a selection of variants, such as to prioritised by the multi-trait p-value, to eliminate false discoveries, quantify and dissect pleiotropy. We believe that the first feature is already well illustrated and quantified as FDRed has been compared side-by-side with the conventional FDR calculated based on p-values in Figure 2, the 9th and 13th paragraphs of Results. As discussed in the 2nd and 3rd paragraphs of the revised Discussion, ‘The FDRed estimated from the sign agreement between results from bulls and cows is smaller than the conventional FDR when variants are chosen that pass a significance test in both sexes’.

The second feature of EDME is an improvement on our previous methods where only the multi-trait p-value for pleiotropy was estimated with uncorrelated traits. The quantified pleiotropic features at the variant level in the current study, e.g., the number of traits associated with each variant, were not available in our previous methods. We have expanded

the discussion around this point in the 4th paragraph in the revised Discussion. Jordan et al 2019 [6] is the most recent study of pleiotropy in uncorrelated traits, therefore, we focus the discussion of their work with ours in the 1st, 4th and 5th paragraphs in the revised Discussion. Briefly, both Jordan et al 2019 [6] (and also other cited analysis with correlated traits) and we reached the same conclusion that there is widespread pleiotropy for uncorrelated traits. Our analysis starts with trait decorrelation and single-trait GWAS and focuses on finding pleiotropic variants in general, while Jordan et al 2019 starts with pre-existed GWAS results and decorrelated the variant-trait association matrix and focused on horizontal pleiotropy. Jordan et al 2019 quantified the magnitude and number of associated traits for all variants that entered the study while our analysis focused on a set of pleiotropic variants prioritised by the multi-trait p-value to dissect pleiotropy using the observed (True Effects) TE in the 900,820 true combinations of trait by variant. We believe that the two analysis have analytical features suitable for different purposes and types of data availability in two species.

2. I find the Cholesky decorrelation method very confusing, and practically no explanation of it is provided in this manuscript. I understand in general terms that the idea is to guarantee that the traits being measured have no overall correlation with each other, but I do would like to see some discussion of why this is important to the analysis. Additionally, I do not understand why the traits are ordered sequentially, rather than being compiled into a single matrix and decorrelated all at once, or why ordering by missingness is appropriate. This is an extremely important methodological detail, because it seems to me that the entire interpretation of the analysis is dependent on the ordering that is chosen. For example, the authors point out in the section "Trait relationships informed by Cholesky decorrelation" that a variant that affects protein yield and survival by corresponding amounts would be treated as having an effect on protein yield and no effect on survival. As a result, if that same variant had an effect on a trait later in the list, such as center ligament (trait 11), that would be counted as a shared variant between protein yield and center ligament, but not between survival and center ligament. How much do the results and interpretation of the analysis change if the order of traits is permuted? Wouldn't it make more sense to use some kind of Mendelian randomization or mediation analysis to determine which traits to assign effects to, rather than assuming that the trait with the least missing data is the true causal trait?

Author response: We apologise for the lack of a detailed description of the Cholesky-transformation. In the revised manuscript we have significantly expanded the discussion

around this analysis. More descriptions are added to the second paragraph in Results ‘*Single-trait GWAS and conventional multi-trait meta-analysis in bull and cow populations*’ where Cholesky transformation is explained. A new Supplementary Figure S1 was added to further explain the motivation and interpretation of the Cholesky-transformed traits. A new paragraph (the current 1st paragraph in Discussion) has been added to the Discussion to discuss the usage of the Cholesky transformation. More details are also added to the Methods (3rd paragraph of Methods). To summarise, the Cholesky transformation decorrelated all traits at once. It can be described by saying that the Kth Cholesky transformed trait can be interpreted as the Kth original trait corrected for the preceding K-1 traits. By ordering the traits in a fashion that the traits with most complete data come first, we make maximal use of the data. This is because all animals recorded for the Kth trait can be used to calculate the Kth Cholesky transformed trait, as this calculation only requires data from the preceding K-1 traits and animals with the Kth trait recorded have data on the K-1 preceding traits. If we had complete data for 34 traits matched in bulls and cows (e.g., our 2017 study) then we could freely consider the order of the Cholesky-transformation. However, this was not the case and having 34 traits with complete records allowing for free consideration of the trait order would result in the sample size for all traits being determined by the trait with the smallest number of records, i.e., 1439 bulls and 4086 cows (instead of up to 11923 bulls and up to 32347 cows).

Most of the results are not dependent on this ordering of the traits. For instance, it does not affect the estimates of FDR or the number of traits affected by a variant. A disadvantage of any transformation is that the transformed traits are hard to interpret biologically. However, this applies to any transformation including that used by Jordan et al. The Cholesky traits can at least be described in simple language as the Kth trait corrected for the preceding K-1 traits. It is not our major purpose to distinguish between horizontal and vertical pleiotropy. However, the Cholesky traits have a natural interpretation with regard to vertical pleiotropy or Mendelian randomization in the case where the causal trait precedes targeted traits and we wanted to point this out. However, as we state in the paper, there is limited power to distinguish cause and effect.

We have added statements ‘We also note that alterations to trait order of the Cholesky-transformation, which is determined by the sample size across traits and sexes, would change the interpretation of CT traits hence impact on the interpretation of the CT trait relationships. As the traits were ordered from large to small sample size, discovered putative causal trait

relationships would be biased towards traits with large sample size to cause traits with small sample size.’ to the 7th paragraph of Discussion where the trait relationships as supported by Mendelian Ransommisation analysis were discussed. Because the major focus of the current study is to identify pleiotropic variants associated with uncorrelated traits with the effect-direction agreement, ‘A more systematic analysis of raw and CT trait relationships aiming at understanding biology between specific trait groups with larger sample size is required in the future’.

3. The authors argue that the ability to study pleiotropic effects of variants is novel, and go so far as to state in the Discussion that the kind of widespread pleiotropy they observe here has never before been reported in mammals. As I mentioned above, I primarily work in humans, but in the field of human genetics I am aware of at least three papers published in the last year that report widespread pleiotropy, some of them using similar methods to this manuscript: Verbanck et al. 2018 (<https://www.nature.com/articles/s41588-018-0099-7>), Watanabe et al. 2019 (<https://www.nature.com/articles/s41588-019-0481-0>), and Jordan et al. 2019 (<https://genomebiology.biomedcentral.com/articles/10.1186/s13059-019-1844-7>). (In the interest of full disclosure, I am an author on one of these papers.) A great deal has been written about the presence of pleiotropy in GWAS traits, and much of it in the last few years has been highlighting the pervasiveness of pleiotropy and what that implies about the genetic architecture of traits. Using a multi-trait GWAS meta analysis to identify widespread pleiotropy in a mammalian species is not novel in itself. That is not to say that this manuscript does not contain novel results with interesting implications for the field, but more discussion of the context and the current state of the field is required.

Author response: We agree with the reviewer that the identification of widespread pleiotropy in mammals is not novel. We have revised these statements in the current 5th paragraph of the Discussion. Specifically, these statements are revised as ‘These results support the conclusion that pleiotropic effects on uncorrelated traits are widespread. These conclusions are also reached by Jordan et al 2019 [6] using a different analysis in humans. Widespread pleiotropy was also found by analyses of GWAS of correlated traits in humans [7, 8].’. But our findings, such as the unevenly distributed pleiotropic effects and that a very small proportion of variants displaying extreme pleiotropy appear to be novel. Therefore, the last sentence of this paragraph is revised as ‘To our knowledge, these findings of pleiotropy are unique to the current study’.

In addition to these three major issues, there are a large number of typos, grammatical errors, and poor formatting that make the manuscript fairly difficult to follow in places. I am not going to go through the manuscript in detail to find all these errors, because a peer reviewer should not have to be a copy editor, but please proofread carefully and make sure all the figures and tables have appropriate legends, labels, and captions.

Author response: We apologise for these errors. We have carefully proof-read the manuscript to eliminate these errors.

References:

1. Bycroft C, Freeman C, Petkova D, Band G, Elliott LT, Sharp K, et al. The UK Biobank resource with deep phenotyping and genomic data. *Nature*. 2018;562(7726):203.
2. Buniello A, MacArthur JAL, Cerezo M, Harris LW, Hayhurst J, Malangone C, et al. The NHGRI-EBI GWAS Catalog of published genome-wide association studies, targeted arrays and summary statistics 2019. *Nucleic acids research*. 2018;47(D1):D1005-D12.
3. Flint J, Eskin E. Genome-wide association studies in mice. *Nature Reviews Genetics*. 2012;13(11):807.
4. Bowden J, Davey Smith G, Burgess S. Mendelian randomization with invalid instruments: effect estimation and bias detection through Egger regression. *International journal of epidemiology*. 2015;44(2):512-25.
5. Zhu Z, Zheng Z, Zhang F, Wu Y, Trzaskowski M, Maier R, et al. Causal associations between risk factors and common diseases inferred from GWAS summary data. *Nature communications*. 2018;9(1):224.
6. Jordan DM, Verbanck M, Do R. HOPS: a quantitative score reveals pervasive horizontal pleiotropy in human genetic variation is driven by extreme polygenicity of human traits and diseases. *Genome Biology*. 2019;20(1):222. doi: 10.1186/s13059-019-1844-7.
7. Verbanck M, Chen C-Y, Neale B, Do R. Detection of widespread horizontal pleiotropy in causal relationships inferred from Mendelian randomization between complex traits and diseases. *Nature genetics*. 2018;50(5):693.
8. Watanabe K, Stringer S, Frei O, Umićević Mirkov M, de Leeuw C, Polderman TJC, et al. A global overview of pleiotropy and genetic architecture in complex traits. *Nature Genetics*. 2019;51(9):1339-48. doi: 10.1038/s41588-019-0481-0.

REVIEWERS' COMMENTS:

Reviewer #1 (Remarks to the Author):

Thank you to the authors for addressing my questions. The changes look fine to me.

Reviewer #3 (Remarks to the Author)

The authors have substantially revised the manuscript, and as a result addressed most of my comments very well, including my concerns about motivation, prior work, and methodological details of the Cholesky decomposition method. I greatly appreciate the large amount of work that went into this revision.

There are still two issues that I feel have not been adequately addressed:

1. The authors assert in their response to my comments that there is ample comparison between FDRed and traditional p-value based FDR in the manuscript; I disagree. The comparison they point to, in Figure 2, shows only that FDRed is consistently lower than traditional FDR and therefore identifies more potentially pleiotropic variants. However, I cannot tell from these comparisons whether the FDRed is a more accurate model of the underlying data than the traditional FDR. To put it another way, there is a large class of variants that have FDRed much lower than traditional FDR. Are those variants truly functional variants that we can now identify because the FDRed method has improved sensitivity? Or are they low-confidence variants that the traditional FDR method was screening out and the FDRed method inappropriately classes as high-confidence? Figure 2 shows that the methods are **different**, but does not show which is **better**.

2. I am unsatisfied with the answer to the comments from both me and Reviewer 1 about the interpretation of Cholesky-transformed traits. The authors' response to my comments and the corresponding additions to the Discussion section point out that any transformation leaves difficulty in interpreting the transformed traits, and we must always be cautious in interpreting the cause and effect after these transformations, which I agree with. They also assert that interpreting specific effects is not their goal, and most of their results are not changed by the order of traits. However, a very large portion of the manuscript is devoted to discussing the relationships between specific CT traits: the sections labeled "The genome-wide characteristics of multi-trait FDR and pleiotropy," "Characteristics of trait-related variants," and "Novel pleiotropic variants" all have extensive discussion of specific CT traits, and use the associations of variants with specific CT traits to draw conclusions about biology, as though the CT traits straightforwardly correspond to their raw counterparts. For example, the authors identify the FTO locus as having effects on specific CT traits including fat percentage, milk production, and stature, and use these specific traits and the way they change when co-analyzed with neighboring genes to draw a parallel with human FTO and its effects on adiposity. If the authors take seriously the idea that there is no straightforward interpretation of transformed traits, this kind of interpretation of individual trait effects should either be done exclusively using raw traits, or be accompanied by some kind of sensitivity analysis showing that the interpretation is not changed by the exact transformation scheme used.

Letter of response to referees

Revised MS# COMMSBIO-19-1181-T, 'Effect direction meta-analysis of GWAS identifies extreme, prevalent and shared pleiotropy in a large mammal'

We thank the reviewers for their effort and time in providing feedback on the manuscript. We have revised our manuscript accordingly. Please find the detailed point-by-point response to the reviewers in the following text. The author's response is in blue text.

REVIEWERS' COMMENTS:

Reviewer #1 (Remarks to the Author):

Thank you to the authors for addressing my questions. The changes look fine to me.

Author Response: we thank for reviewer for considering our manuscript revision.

Reviewer # 3 (Remarks to the Author)

The authors have substantially revised the manuscript, and as a result addressed most of my comments very well, including my concerns about motivation, prior work, and methodological details of the Cholesky decomposition method. I greatly appreciate the large amount of work that went into this revision.

There are still two issues that I feel have not been adequately addressed:

1. The authors assert in their response to my comments that there is ample comparison between FDRed and traditional p-value based FDR in the manuscript; I disagree. The comparison they point to, in Figure 2, shows only that FDRed is consistently lower than traditional FDR and therefore identifies more potentially pleiotropic variants. However, I cannot tell from these comparisons whether the FDRed is a more accurate model of the

underlying data than the traditional FDR. To put it another way, there is a large class of variants that have FDRed much lower than traditional FDR. Are those variants truly functional variants that we can now identify because the FDRed method has improved sensitivity? Or are they low-confidence variants that the traditional FDR method was screening out and the FDRed method inappropriately classes as high-confidence? Figure 2 shows that the methods are *different*, but does not show which is *better*.

Author response: we thank the reviewer for considering our revised manuscript and providing us with feedback. FDRed has advantages over the conventional FDR which we hope we have adequately explained. The conventional FDR is dependent on the p-values which in turn are dependent on the statistical model matching the real data. The FDRed is an empirical estimate, that is, it is based on the ability to confirm results in independent data. In fact, when the p-values in a single population (bull or cow) are used to select significant variants, FDRed is higher (i.e. more conservative) than the conventional FDR. When variants are selected that are significant in both populations the FDRed is lower than the conventional FDR for variants selected in only one population. Thus, by selecting variants that are significant in both bulls and cows at a modest $p < 10^{-6}$ we obtain a set with a very low FDR based on an empirical and conservative FDR. GWAS find variants associated with the trait but they are not necessarily causal. Nevertheless, we have tested the enrichment of conserved sites and expression QTLs [1, 2] in GWAS hits at a set of p-value thresholds from lenient to stringent imposed in bulls and cows. As shown in Supplementary Note 1, at a relatively lenient p-value threshold such as 5×10^{-4} , there was already a significant enrichment of functional and evolutionary signals in GWAS. At the p-value threshold of 5×10^{-4} , the FDRed was < 0.01 for both the single- and multi-trait analysis. As mentioned in the text, another advantage of the FDRed logic is that we can use it to estimate the number of traits affected by variants. We hope that the FDRed may be useful to other scientists who carry out a meta-analysis of published summary statistics from GWAS.

2. I am unsatisfied with the answer to the comments from both me and Reviewer 1 about the interpretation of Cholesky-transformed traits. The authors' response to my comments and the corresponding additions to the Discussion section point out that any transformation leaves difficulty in interpreting the transformed traits, and we must always be cautious in

interpreting the cause and effect after these transformations, which I agree with. They also assert that interpreting specific effects is not their goal, and most of their results are not changed by the order of traits. However, a very large portion of the manuscript is devoted to discussing the relationships between specific CT traits: the sections labeled "The genome-wide characteristics of multi-trait FDR and pleiotropy," "Characteristics of trait-related variants," and "Novel pleiotropic variants" all have extensive discussion of specific CT traits, and use the associations of variants with specific CT traits to draw conclusions about biology, as though the CT traits straightforwardly correspond to their raw counterparts. For example, the authors identify the FTO locus as having effects on specific CT traits including fat percentage, milk production, and stature, and use these specific traits and the way they change when co-analyzed with neighboring genes to draw a parallel with human FTO and its effects on adiposity. If the authors take seriously the idea that there is no straightforward interpretation of transformed traits, this kind of interpretation of individual trait effects should either be done exclusively using raw traits, or be accompanied by some kind of sensitivity analysis showing that the interpretation is not changed by the exact transformation scheme used.

Author response: We used the Cholesky transformation because we wished to assess the degree of pleiotropy among uncorrelated traits. Although we agree with the reviewer that the Cholesky traits are difficult to interpret, we still think that this does not affect the majority of our conclusions, including the results 'X loci is associated with Y number of CT (uncorrelated) traits'. In fact, finding variants/loci associated with uncorrelated traits is our primary goal of the study. Nevertheless, to address the comments from the reviewer, we have made the following changes in the text:

After carefully went through the text, we moved those sections where we described specific Cholesky traits and the related discussions into Supplementary Note 2. We retained those texts where we only described the number of associated CT traits for noted variants or loci. For the section 'Characteristics of trait-related variants', we added a statement that 'However, we note that these variants were related to CT traits and the interpretation of them is different from the raw traits.' For the results related to FTO and KRT loci as shown in Fig 6, we have re-done the joint analysis using results of 37 GWAS of raw traits and we have replaced the heatmap for the CT traits with the heatmap for the raw traits in the new Fig 6. Related texts in the Results and Discussion are also changed accordingly. Overall, for these two loci, the results regarding the variant effects on CT and raw traits are similar.

1. Xiang R, Hayes BJ, Vander Jagt CJ, MacLeod IM, Khansefid M, Bowman PJ, et al. Genome variants associated with RNA splicing variations in bovine are extensively shared between tissues. *BMC Genomics*. 2018;19(1):521. doi: 10.1186/s12864-018-4902-8.
2. Xiang R, Berg Ivd, MacLeod IM, Hayes BJ, Prowse-Wilkins CP, Wang M, et al. Quantifying the contribution of sequence variants with regulatory and evolutionary significance to 34 bovine complex traits. *Proceedings of the National Academy of Sciences*. 2019;116(39):19398-408. doi: 10.1073/pnas.1904159116.